# Latent Neural ODEs with Sparse Bayesian Multiple Shooting

**Valerii Iakovlev,**\* **Cagatay Yildiz,**† **Markus Heinonen,**\* **Harri Lähdesmäki**\*

## Abstract

Training dynamic models, such as neural ODEs, on long trajectories is a hard problem that requires using various tricks, such as trajectory splitting, to make model training work in practice. These methods are often heuristics with poor theoretical justifications, and require iterative manual tuning. We propose a principled multiple shooting technique for neural ODEs that splits the trajectories into manageable short segments, which are optimised in parallel, while ensuring probabilistic control on continuity over consecutive segments. We derive variational inference for our shooting-based latent neural ODE models and propose amortized encodings of irregularly sampled trajectories with a transformer-based recognition network with temporal attention and relative positional encoding. We demonstrate efficient and stable training, and state-of-the-art performance on multiple large-scale benchmark datasets.

## 1 Introduction

Dynamical systems, from biological cells to weather, evolve according to their underlying mechanisms, often described by differential equations. In data-driven system identification we aim to learn the rules governing a dynamical system by observing the system for a time interval $[0, T]$, and fitting a model of the underlying dynamics to the observations by gradient descent. Such optimisation suffers from the *curse of length*: complexity of the loss function grows with the length of the observed trajectory (Ribeiro et al., 2020). For even moderate $T$ the loss landscape can become highly complex and gradient descent fails to produce a good fit (Metz et al., 2021). To alleviate this problem previous works resort to cumbersome heuristics, such as iterative training and trajectory splitting (Yildiz et al., 2019; Kochkov et al., 2021; HAN et al., 2022; Lienen & Günnemann, 2022).

The optimal control literature has a long history of multiple shooting methods, where the trajectory fitting is split into piecewise segments that are easy to optimise, with constraints to ensure continuity across the segments (van Domselaar & Hemker, 1975; Bock & Plitt, 1984; Baake et al., 1992). Multiple-shooting based models have simpler loss landscapes, and are practical to fit by gradient descent (Voss et al., 2004; Heiden et al., 2022; Turan & Jäschke, 2022; Hegde et al., 2022).

Inspired by this line of work, we develop a shooting-based latent neural ODE model (Chen et al., 2018; Rubanova et al., 2019; Yildiz et al., 2019; Massaroli et al., 2020). Our multiple shooting formulation generalizes standard approaches by sparsifying the shooting variables in a probabilistic setting to account for irregularly sampled time grids and redundant shooting variables. We furthermore introduce an attention-based (Vaswani et al., 2017) encoder architecture for latent neural ODEs that is compatible with our sparse shooting formulation and can handle noisy and partially observed high-dimensional data. Consequently, our model produces state-of-the-art results, naturally handles the problem with long observation intervals, and is stable and quick to train. Our contributions are:

- We introduce a latent neural ODE model with quick and stable training on long trajectories.
- We derive sparse Bayesian multiple shooting – a Bayesian version of multiple shooting with efficient utilization of shooting variables and a continuity-inducing prior.
- We introduce a transformer-based encoder with novel time-aware attention and relative positional encodings, which efficiently handles data observed at arbitrary time points.

---

\*Aalto University, Finland. Corresponding author: `valerii.iakovlev@aalto.fi`.
†University of Tübingen, Germany. Code: https://github.com/yakovlev31/msvi

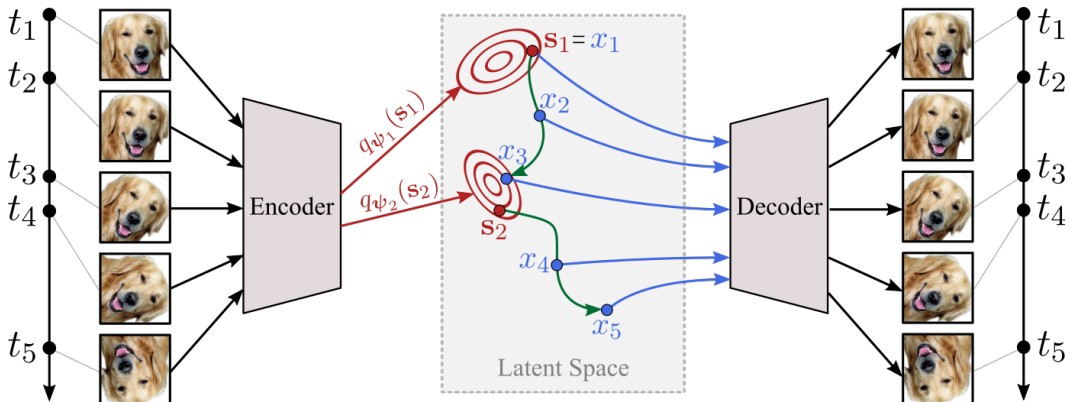

Figure 2: **Method overview** with two blocks (see Section 3.1). The encoder maps the input sequence $\boldsymbol{y}_{1:5}$ observed at arbitrary time points $t_{1:5}$ to two distributions $q_{\boldsymbol{\psi}_1}(\boldsymbol{s}_1), q_{\boldsymbol{\psi}_2}(\boldsymbol{s}_2)$ from which we sample shooting variables $\boldsymbol{s}_1, \boldsymbol{s}_2$. Then, $\boldsymbol{s}_1, \boldsymbol{s}_2$ are used to compute two sub-trajectories that define the latent trajectory $\boldsymbol{x}_{1:5}$ from which the decoder reconstructs the input sequence.

## 2 PROBLEM SETTING AND BACKGROUND

**Data.** We observe a dynamical system at arbitrary consecutive time points $t_{1:N} = (t_1, ..., t_N)$, which generates an observed trajectory $\boldsymbol{y}_{1:N} = (\boldsymbol{y}_1, ..., \boldsymbol{y}_N)$, where $\boldsymbol{y}_i := \boldsymbol{y}(t_i) \in \mathbb{R}^D$. Our goal is to model the observations and forecast the future states. For brevity we present our methodology for a single trajectory, but extension to many trajectories is straightforward.

**Latent Neural ODE models.** L-NODE models (Chen et al., 2018; Rubanova et al., 2019) relate the observations $\boldsymbol{y}_{1:N}$ to a latent trajectory $\boldsymbol{x}_{1:N} := (\boldsymbol{x}_1, ..., \boldsymbol{x}_N)$, where $\boldsymbol{x}_i := \boldsymbol{x}(t_i) \in \mathbb{R}^d$, and learn dynamics in the latent space. An L-NODE model is defined as:

$$\boldsymbol{x}_i = \text{ODEsolve}(\boldsymbol{x}_1, t_1, t_i, f_{\theta_{\text{dyn}}}), \qquad i = 2, ..., N, \quad (1)$$

$$\boldsymbol{y}_i | \boldsymbol{x}_i \sim p(\boldsymbol{y}_i | g_{\theta_{\text{dec}}}(\boldsymbol{x}_i)), \qquad i = 1, ..., N. \quad (2)$$

Variable $\boldsymbol{x}_1$ is the initial state at time $t_1$. Dynamics function $f_{\theta_{\text{dyn}}}$ is the time derivative of $\boldsymbol{x}(t)$, and $\text{ODEsolve}(\boldsymbol{x}_1, t_1, t_i, f_{\theta_{\text{dyn}}})$ is defined as the solution of the following initial value problem at time $t_i$:

$$\frac{d\boldsymbol{x}(t)}{dt} = f_{\theta_{\text{dyn}}}(t, \boldsymbol{x}(t)), \quad \boldsymbol{x}(t_1) = \boldsymbol{x}_1, \quad t \in [t_1, t_i]. \quad (3)$$

Decoder $g_{\theta_{\text{dec}}}$ maps the latent state $\boldsymbol{x}_i$ to the parameters of $p(\boldsymbol{y}_i | g_{\theta_{\text{dec}}}(\boldsymbol{x}_i))$. Dynamics and decoder functions are neural networks with parameters $\theta_{\text{dyn}}$ and $\theta_{\text{dec}}$. In typical applications, data is high-dimensional whereas the dynamics are modeled in a low-dimensional latent space, i.e., $d \ll D$.

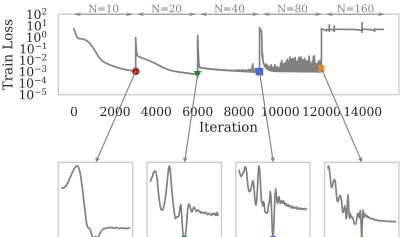

Figure 1: **Top:** Train loss of L-NODE model using iterative training heuristic. We start training on a short trajectory ($N = 10$), and double its length every 3000 iterations. The training fails for the longest trajectory. **Bottom:** 1-D projection of the loss landscape around the parameters to which the optimizer converged for a given trajectory length. Complexity of the loss grows dramatically with $N$.

L-NODE models are commonly trained by minimizing a loss function, e.g., evidence lower bound (ELBO), via gradient descent (Chen et al., 2018; Yildiz et al., 2019). In gradient-based optimization complexity of the loss landscape plays a crucial role in the success of the optimization. However, it has been empirically shown that the loss landscape of L-NODE-like models (i.e., models that compute latent trajectory $\boldsymbol{x}_{1:N}$ from initial state $\boldsymbol{x}_1$) is strongly affected by the length of the simulation interval $[t_1, t_N]$ (Voss et al., 2004; Metz et al., 2021; Heiden et al., 2022). Furthermore, Ribeiro et al. (2020) show that the loss complexity in terms of Lipschitz constant can grow exponentially with the length of $[t_1, t_N]$. Figure 1 shows an example of this phenomenon (details in Appendix A).

## 3    METHODS

In Section 3.1, we present our latent neural ODE formulation that addresses the curse of length by sparse multiple shooting. In Section 3.2 we describe the generative model, inference, and forecasting procedures. In Section 3.3 we describe our time-aware, attention-based encoder architecture that complements our sparse multiple shooting framework.

### 3.1    LATENT NEURAL ODES WITH SPARSE MULTIPLE SHOOTING

**Multiple shooting.**    A simple and effective method for solving optimisation problems with long simulation intervals is to split these intervals into short, non-overlapping sub-intervals that are optimised in parallel. This is the main idea of a technique called multiple shooting (Hemker, 1974; Bock & Plitt, 1984). To apply multiple shooting to an L-NODE model we introduce new parameters, called shooting variables, $\boldsymbol{s}_{1:N-1} = (\boldsymbol{s}_1, \ldots, \boldsymbol{s}_{N-1})$ with $\boldsymbol{s}_i \in \mathbb{R}^d$ that correspond to time points $t_{1:N-1}$, and redefine the L-NODE model as

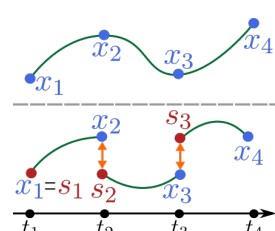

$$\boldsymbol{x}_1 = \boldsymbol{s}_1, \tag{4}$$
$$\boldsymbol{x}_i = \mathrm{ODEsolve}(\boldsymbol{s}_{i-1}, t_{i-1}, t_i, f_{\theta_{\mathrm{dyn}}}), \tag{5}$$
$$\boldsymbol{y}_i | \boldsymbol{x}_i \sim p\big(\boldsymbol{y}_i | g_{\theta_{\mathrm{dec}}}(\boldsymbol{x}_i)\big). \tag{6}$$

Figure 3: **Top**: Trajectory over $[t_1, t_4]$, $\boldsymbol{x}_i$ is computed from $\boldsymbol{x}_1$. **Bottom**: $[t_1, t_4]$ is split into three sub-intervals, $\boldsymbol{x}_i$ is computed from $\boldsymbol{s}_{i-1}$.

The initial state $\boldsymbol{x}_1$ is represented by the first shooting variable $\boldsymbol{s}_1$, and the latent state $\boldsymbol{x}_i$ is computed from the previous shooting variable $\boldsymbol{s}_{i-1}$. This gives short simulation intervals $[t_{i-1}, t_i]$, which greatly reduces complexity of the loss landscape. Continuity of the entire piecewise trajectory is enforced via constraints on the distances between $\boldsymbol{x}_i$ and $\boldsymbol{s}_i$ (see Figure 3), which we discuss in Section 3.2. Multiple shooting leads to a new optimisation problem over $\theta_{\mathrm{dyn}}, \theta_{\mathrm{dec}}$, and $\boldsymbol{s}_{1:N-1}$.

**Sparse multiple shooting.**    Multiple shooting assigns a shooting variable to every time point (see Figure 3). For irregular or densely sampled time grids this approach might result in redundant shooting variables and excessively short and uninformative sub-intervals due to high concentration of time points in some regions of the time grid.

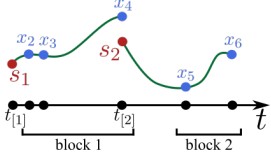

We propose to fix these problems by sparsifying the shooting variables. Instead of assigning a shooting variable to every time point, we divide the time grid into $B$ non-overlapping *blocks* and assign a single shooting variable to each block. For block $b \in \{1, ..., B\}$, we define an index set $\mathcal{I}_b$ containing indices of consecutive time points associated with that block such that $\cup_b \mathcal{I}_b = \{2, \ldots, N\}$. We do not include the first time point $t_1$ in any of the blocks. With every block $b$ we associate observations $\{\boldsymbol{y}_i\}_{i \in \mathcal{I}_b}$, time points $\{t_i\}_{i \in \mathcal{I}_b}$ and a shooting variable $\boldsymbol{s}_b$ placed at the first time point before the block. The temporal position of $\boldsymbol{s}_b$ is denoted by $t_{[b]}$. Latent states $\{\boldsymbol{x}_i\}_{i \in \mathcal{I}_b}$ are computed from $\boldsymbol{s}_b$ as

Figure 4:    An example of sparse multiple shooting with $B = 2$, $\mathcal{I}_1 = \{2, 3, 4\}$ and $\mathcal{I}_2 = \{5, 6\}$.

$$\boldsymbol{x}_i = \mathrm{ODEsolve}(\boldsymbol{s}_b, t_{[b]}, t_i, f_{\theta_{\mathrm{dyn}}}), \quad i \in \mathcal{I}_b. \tag{7}$$

As shown in Figure 4, this approach reduces the number of shooting variables and grants finer control over the length of each sub-interval to ensure that it is both sufficiently long to contain enough dynamics information and sufficiently short to keep the loss landscape not too complex.

As illustrated in Figure 4, an ODE solution (Eq. 7) does not necessarily match the corresponding shooting variable. Standard multiple shooting formulations enforce continuity of the entire trajectory via a hard constraint or a penalty term (Voss et al., 2004; Jordana et al., 2021; Turan & Jäschke, 2022). Instead, we propose to utilize Bayesian inference and naturally encode continuity as a prior which leads to sparse Bayesian multiple shooting which we discuss in the next section.

## 3.2 MODEL, INFERENCE, AND FORECASTING

**Model.** Our model is a latent neural ODE with sparse multiple shooting (Section 3.1). To infer the parameters $s_{1:B}, \theta_{\text{dyn}}$, and $\theta_{\text{dyn}}$ we use Bayesian inference with the following prior:

$$p(s_{1:B}, \theta_{\text{dyn}}, \theta_{\text{dec}}) = p(s_{1:B}|\theta_{\text{dyn}})p(\theta_{\text{dyn}})p(\theta_{\text{dec}}), \tag{8}$$

where $p(\theta_{\text{dyn}}), p(\theta_{\text{dec}})$ are Gaussians, and the *continuity inducing prior* $p(s_{1:B}|\theta_{\text{dyn}})$ is defined as

$$p(s_{1:B}|\theta_{\text{dyn}}) = p(s_1)\prod_{b=2}^{B} p(s_b|s_{b-1}, \theta_{\text{dyn}}) = p(s_1)\prod_{b=2}^{B} \mathcal{N}\left(s_b|\text{ODEsolve}(s_{b-1}, t_{[b-1]}, t_{[b]}, f_{\theta_{\text{dyn}}}), \sigma_c^2 I\right), \tag{9}$$

where $p(s_1)$ is a diagonal Gaussian, $\mathcal{N}$ is the Gaussian distribution, $I \in \mathbb{R}^{d \times d}$ is identity matrix, and parameter $\sigma_c^2$ controls the strength of the prior. The continuity prior forces the shooting variable $s_b$ and the final state of the previous block $b-1$, which is obtained using the dynamics model, to be close (e.g., $s_2$ and $x(t_{[2]}) = x_4$ in Fig. 4), thus promoting continuity of the entire trajectory.

With the priors above, we get the following generative model

$$\theta_{\text{dyn}}, \theta_{\text{dec}} \sim p(\theta_{\text{dyn}})p(\theta_{\text{dec}}), \quad s_{1:B}|\theta_{\text{dyn}} \sim p(s_{1:B}|\theta_{\text{dyn}}), \tag{10}$$

$$x_1 = s_1, \tag{11}$$

$$x_i = \text{ODEsolve}(s_b, t_{[b]}, t_i, f_{\theta_{\text{dyn}}}), \qquad\qquad b \in \{1, ..., B\},\ i \in \mathcal{I}_b, \tag{12}$$

$$y_i|x_i \sim p(y_i|g_{\theta_{\text{dec}}}(x_i)), \qquad\qquad i = 1, ..., N. \tag{13}$$

Since $x_{1:N}$ are deterministic functions of $s_{1:B}$ and $\theta_{\text{dyn}}$, we have the following joint distribution (see Appendix B for more details)

$$p(y_{1:N}, s_{1:B}, \theta_{\text{dyn}}, \theta_{\text{dec}}) = p(y_{1:N}|s_{1:B}, \theta_{\text{dyn}}, \theta_{\text{dec}})p(s_{1:B}|\theta_{\text{dyn}})p(\theta_{\text{dyn}})p(\theta_{\text{dec}}). \tag{14}$$

**Inference.** We use variational inference (Blei et al., 2017) to approximate the true posterior $p(\theta_{\text{dyn}}, \theta_{\text{dec}}, s_{1:B}|y_{1:N})$ by an approximate posterior

$$q(\theta_{\text{dyn}}, \theta_{\text{dec}}, s_{1:B}) = q(\theta_{\text{dyn}})q(\theta_{\text{dec}})q(s_{1:B}) = q_{\psi_{\text{dyn}}}(\theta_{\text{dyn}})q_{\psi_{\text{dec}}}(\theta_{\text{dec}})\prod_{b=1}^{B} q_{\psi_b}(s_b) \tag{15}$$

with variational parameters $\psi_{\text{dyn}}, \psi_{\text{dec}}$, and $\psi_{1:B} = (\psi_1, \ldots, \psi_B)$. Note that contrary to standard VAEs, which use point estimates of $\theta_{\text{dyn}}$ and $\theta_{\text{dec}}$, we extent the variational inference to these parameters to adequately handle the uncertainty. To avoid direct optimization over the local variational parameters $\psi_{1:B}$, we use amortized variational inference (Kingma & Welling, 2013) and learn an encoder $h_{\theta_{\text{enc}}}$ with parameters $\theta_{\text{enc}}$ which maps observations $y_{1:N}$ to $\psi_{1:B}$ (see Section 3.3). We denote the amortized shooting distributions $q_{\psi_b}(s_b|y_{1:N}, \theta_{\text{enc}})$, where $\psi_b = h_{\theta_{\text{enc}}}(y_{1:N})$, simply as $q(s_b)$ or $q_{\psi_b}(s_b)$ for brevity. We assume $q_{\psi_{\text{dyn}}}, q_{\psi_{\text{dec}}}$, and $q_{\psi_b}$ to be diagonal Gaussians.

With a fully factorised $q(s_{1:B})$ we can sample the shooting variables $s_{1:B}$ independently which allows to compute the latent states $x_{1:N}$ in parallel by simulating the dynamics only over short subintervals. If the posterior $q(s_{1:B})$ followed the structure of the prior $p(s_{1:B}|\theta_{\text{dyn}})$ we would not be able to utilize these benefits of multiple shooting since to sample $s_{1:B}$ we would need to simulate the whole trajectory $s_{1:B}$ starting at $s_1$.

In variational inference we minimize the Kullback-Leibler divergence between the variational approximation and the true posterior,

$$\text{KL}\left[q(\theta_{\text{dyn}}, \theta_{\text{dec}}, s_{1:B})\|p(\theta_{\text{dyn}}, \theta_{\text{dec}}, s_{1:B}|y_{1:N})\right], \tag{16}$$

which is equivalent to maximizing the ELBO which for our model is defined as

$$\mathcal{L} = \underbrace{\mathbb{E}_{q(\theta_{\text{dec}}, s_1)}\left[\log p(y_1|s_1, \theta_{\text{dec}})\right]}_{\textit{(i) data likelihood}} + \sum_{b=1}^{B}\sum_{i \in \mathcal{I}_b} \underbrace{\mathbb{E}_{q(\theta_{\text{dyn}}, \theta_{\text{dec}}, s_b)}\left[\log p(y_i|s_b, \theta_{\text{dyn}}, \theta_{\text{dec}})\right]}_{\textit{(ii) data likelihood}} \tag{17}$$

$$- \underbrace{\text{KL}\left[q(s_1)\|p(s_1)\right]}_{\textit{(iii) initial state prior}} - \sum_{b=2}^{B} \underbrace{\mathbb{E}_{q(\theta_{\text{dyn}}, s_{b-1})}\left[\text{KL}\left[q(s_b)\|p(s_b|s_{b-1}, \theta_{\text{dyn}})\right]\right]}_{\textit{(iv) continuity prior}} \tag{18}$$

$$- \underbrace{\text{KL}\left[q(\theta_{\text{dyn}})\|p(\theta_{\text{dyn}})\right]}_{\textit{(v) dynamics prior}} - \underbrace{\text{KL}\left[q(\theta_{\text{dec}})\|p(\theta_{\text{dec}})\right]}_{\textit{(vi) decoder prior}}. \tag{19}$$

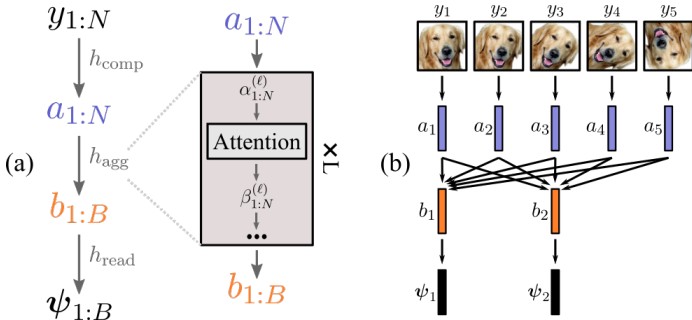

Figure 5: (a) Encoder structure. (b) Encoder with two blocks (i.e., $B = 2$) operating on input sequence $\boldsymbol{y}_{1:5}$ with shooting variables $\boldsymbol{s}_1, \boldsymbol{s}_2$ located at $t_1, t_3$.

Appendix B contains detailed derivation of the ELBO, and fully specifies the model and the approximate posterior. While terms *(iii)*, *(v)* and *(vi)* have a closed form, computation of terms *(i)*, *(ii)* and *(iv)* involves approximations: Monte Carlo sampling for the expectations, and numerical integration for the solution of the initial value problems. Appendix C details the computation of ELBO.

**Forecasting.** Given initial observations $\boldsymbol{y}_{1:m}^*$ of a test trajectory at time points $t_{1:m}^*$ we predict the future observation $\boldsymbol{y}_n^*$ at a time point $t_n^* > t_m^*$ as the expected value of the approximate posterior predictive distribution

$$p(\boldsymbol{y}_n^*|\boldsymbol{y}_{1:m}^*, \boldsymbol{y}_{1:N}) \approx \int p(\boldsymbol{y}_n^*|\boldsymbol{s}_1^*, \theta_{\text{dyn}}, \theta_{\text{dec}}) q_{\boldsymbol{\psi}_1^*}(\boldsymbol{s}_1^*) q_{\boldsymbol{\psi}_{\text{dyn}}}(\theta_{\text{dyn}}) q_{\boldsymbol{\psi}_{\text{dec}}}(\theta_{\text{dec}}) \mathrm{d}\boldsymbol{s}_1^* \mathrm{d}\theta_{\text{dyn}} \mathrm{d}\theta_{\text{dec}}, \qquad (20)$$

where $\boldsymbol{\psi}_1^* = h_{\theta_{\text{enc}}}(\boldsymbol{y}_{1:m}^*)$. The expectation is estimated via Monte Carlo integration (Appendix C). Note that inferring $\boldsymbol{s}_m^*$ instead of $\boldsymbol{s}_1^*$ could lead to more accurate predictions, but in this work we use $\boldsymbol{s}_1^*$ to simplify implementation of the method.

### 3.3 ENCODER

We want to design an encoder capable of operating on irregular time grids, handling noisy and partially observed data, and parallelizing the computation of the local variational parameters $\boldsymbol{\psi}_{1:B}$. Transformer (Vaswani et al., 2017) satisfies most of these requirements, but is not directly applicable to our setup. We design a transformer-based encoder with time-aware attention and continuous relative positional encodings. These modifications provide useful inductive biases and allow the encoder to effectively operate on input sequences with a temporal component. The encoder computes $\boldsymbol{\psi}_{1:B}$ with (see Figure 5 (a-b)):

$$\boldsymbol{\psi}_{1:B} = h_{\theta_{\text{enc}}}(\boldsymbol{y}_{1:N}) = h_{\text{read}}(h_{\text{agg}}(h_{\text{comp}}(\boldsymbol{y}_{1:N}))), \qquad (21)$$

where

1. $h_{\text{comp}} : \mathbb{R}^D \to \mathbb{R}^{D_{\text{low}}}$ compresses observations $\boldsymbol{y}_{1:N} \in \mathbb{R}^{D \times N}$ into a low-dimensional sequence $\boldsymbol{a}_{1:N} \in \mathbb{R}^{D_{\text{low}} \times N}$, where $D_{\text{low}} \ll D$.
2. $h_{\text{agg}} : \mathbb{R}^{D_{\text{low}} \times N} \to \mathbb{R}^{D_{\text{low}} \times B}$ aggregates information across $\boldsymbol{a}_{1:N}$ into $\boldsymbol{b}_{1:B} \in \mathbb{R}^{D_{\text{low}} \times B}$, where $\boldsymbol{b}_i$ is located at the temporal position of $\boldsymbol{s}_i$ (Figure 5 (b)).
3. $h_{\text{read}} : \mathbb{R}^{D_{\text{low}}} \to \mathbb{R}^P$ reads the parameters $\boldsymbol{\psi}_{1:B} \in \mathbb{R}^{P \times B}$ from $\boldsymbol{b}_{1:B}$.

Transformations $h_{\text{comp}}$ and $h_{\text{read}}$ are any suitable differentiable functions. Transformation $h_{\text{agg}}$ is a transformer encoder (Vaswani et al., 2017) which is a sequence-to-sequence mapping represented by a stack of $L$ layers (Figure 5 (a)). Each layer $\ell \in \{1, \dots, L\}$ contains a component called attention sub-layer which maps an input sequence $\boldsymbol{\alpha}_{1:N}^{(\ell)} := (\boldsymbol{\alpha}_1^{(\ell)}, \dots, \boldsymbol{\alpha}_N^{(\ell)}) \in \mathbb{R}^{D_{\text{low}} \times N}$ to an output sequence $\boldsymbol{\beta}_{1:N}^{(\ell)} := (\boldsymbol{\beta}_1^{(\ell)}, \dots, \boldsymbol{\beta}_N^{(\ell)}) \in \mathbb{R}^{D_{\text{low}} \times N}$, except for the last layer which maps $\boldsymbol{\alpha}_{1:N}^{(L)}$ to $\boldsymbol{\beta}_{1:B}^{(L)}$ to match the number of shooting variables. For the first layer, $\boldsymbol{\alpha}_{1:N}^{(1)} = \boldsymbol{a}_{1:N}$, and for the last layer, $\boldsymbol{b}_{1:B} = \text{FF}(\boldsymbol{\beta}_{1:B}^{(L)})$, where $\text{FF}(\cdot)$ is a feed-forward network with a residual connection. In the

following, we drop the index $\ell$ for notational simplicity since each layer has the same structure. The attention sub-layer for the standard, scaled dot-product self-attention (assuming a single attention head) is defined using the dot-product ($C_{ij}^{\mathrm{DP}}$), softmax ($C_{ij}$) and weighted average ($\beta_i$) (Vaswani et al., 2017):

$$C_{ij}^{\mathrm{DP}} = \frac{\langle W_Q \boldsymbol{\alpha}_i, W_K \boldsymbol{\alpha}_j \rangle}{\sqrt{D_{\mathrm{low}}}}, \quad C_{ij} = \frac{\exp\left(C_{ij}^{\mathrm{DP}}\right)}{\sum_{k=1}^{N} \exp\left(C_{ik}^{\mathrm{DP}}\right)}, \quad \boldsymbol{\beta}_i = \sum_{j=1}^{N} C_{ij}(W_V \boldsymbol{\alpha}_j), \tag{22}$$

where $W_Q, W_K, W_V \in \mathbb{R}^{D_{\mathrm{low}} \times D_{\mathrm{low}}}$ are learnable layer-specific parameter matrices, and $C \in \mathbb{R}^{N \times N}$ is the attention matrix. This standard formulation of self-attention works poorly on irregularly sampled trajectories (see Section 4). Next, we discuss modifications that we introduce to make it applicable on irregularly sampled data.

**Temporal attention**     Dot product attention has no notion of time hence can attend to arbitrary elements of the input sequence. To make $\boldsymbol{\beta}_i$ dependent mostly on those input elements that are close to $t_i$ we augment the dot-product attention with temporal attention $C_{ij}^{\mathrm{TA}}$ and redefine the attention matrix as

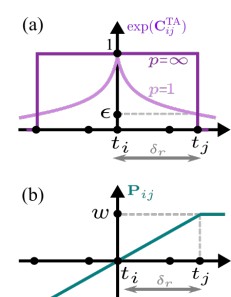

$$C_{ij}^{\mathrm{TA}} = \ln\left(\epsilon\right) \left(\frac{|t_j - t_i|}{\delta_r}\right)^p, \quad C_{ij} = \frac{\exp\left(C_{ij}^{\mathrm{DP}} + C_{ij}^{\mathrm{TA}}\right)}{\sum_{k=1}^{N} \exp\left(C_{ik}^{\mathrm{DP}} + C_{ik}^{\mathrm{TA}}\right)}, \tag{23}$$

where $\epsilon \in (0, 1]$, $p \in \mathbb{N}$ and $\delta_r \in \mathbb{R}_{>0}$ are constants. Since $\exp\left(C_{ij}^{\mathrm{DP}} + C_{ij}^{\mathrm{TA}}\right) = \exp\left(C_{ij}^{\mathrm{DP}}\right)\exp\left(C_{ij}^{\mathrm{TA}}\right)$, the main purpose of temporal attention is to reduce the amount of attention from $\boldsymbol{\beta}_i$ to $\boldsymbol{\alpha}_j$ as the temporal distance $|t_i - t_j|$ grows. Parameter $\delta_r$ defines the distance beyond which $\exp(C_{ij}^{\mathrm{DP}})$ is scaled by at least $\epsilon$, while $p$ defines the shape of the scaling curve. Figure 6 (a) demonstrates shapes of the scaling curves for various values of $p$.

**Relative positional encodings**     To make $\boldsymbol{\beta_i}$ independent of its absolute temporal position $t_i$ we replace the standard global positional encodings with relative positional encodings which we define as

Figure 6: (a) Temporal attention. (b) Relative position encoding.

$$\boldsymbol{P}_{ij} = \boldsymbol{w} \odot \mathrm{hardtanh}\left(\frac{t_j - t_i}{\delta_r}\right), \quad \text{and redefine} \quad \boldsymbol{\beta}_i = \sum_{j=1}^{N} C_{ij}(W_V \boldsymbol{\alpha}_j + \boldsymbol{P}_{ij}), \tag{24}$$

where $\boldsymbol{w} \in \mathbb{R}^d$ is a vector of trainable parameters, $\odot$ is point-wise multiplication, and $\delta_r$ is the same as for temporal attention. This formulation is synergistic with temporal attention as it ensures that $\boldsymbol{\beta}_i$ has useful positional information about $\boldsymbol{\alpha}_j$ only if $|t_i - t_j| < \delta_r$ which further forces $\boldsymbol{\beta}_i$ to depend on input elements close to $t_i$ (see Figure 6 (b)). In this work we share $\boldsymbol{w}$ across attention sub-layers. For further details about the encoder, see Appendix E. In Appendix F we investigate the effects of $p$ and $\delta_r$. In Appendix J we compare our transformer-based aggregation function with ODE-RNN of Rubanova et al. (2019).

Note that our encoder can process input sequences of varying lengths. Also, as discussed in Section 3.2, at test time we set $B = 1$ so that the encoder outputs only the first parameter vector $\boldsymbol{\psi}_1$ since we are only interested in the initial state $\boldsymbol{s}_1$ from which we predict the test trajectory.

## 4   EXPERIMENTS

To demonstrate properties and capabilities of our method we use three datasets: PENDULUM, RMNIST, and BOUNCING BALLS, which consist of high-dimensional ($D = 1024$) observations of physical systems evolving over time (Figure 7) and are often used in literature on modeling of dynamical systems. We generate these datasets on regular and irregular time grids. Unless otherwise stated, we use the versions with irregular time grids. See Appendix D for more details.

We train our model for 300000 iterations with Adam optimizer (Kingma & Ba, 2015) and learning rate exponentially decreasing from $3 \cdot 10^{-4}$ to $10^{-5}$. To simulate the dynamics we use an ODE solver from `torchdiffeq` package (Chen et al., 2018) (dopri5 with rtol = atol = $10^{-5}$). We use second-order dynamics and set the latent space dimension $d$ to 32. See Appendix E for detailed description of training/validation/testing setup and model architecture. Error bars are standard errors evaluated with five random seeds. Training is done on a single NVIDIA Tesla V100 GPU.

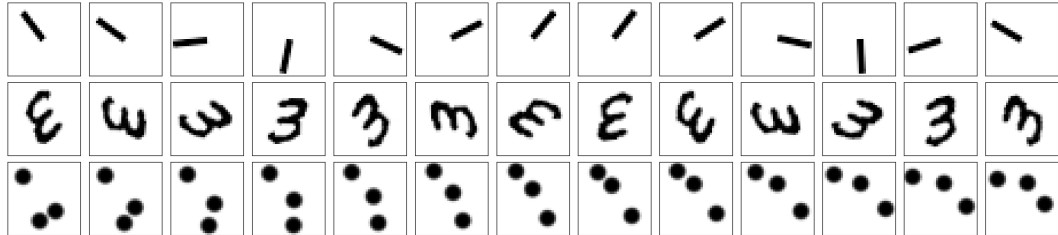

Figure 7: *Top row*: PENDULUM dataset consisting of images of a pendulum moving under the influence of gravity. *Middle row*: RMNIST dataset consisting of images of rotating digits 3. *Bottom row*: BOUNCING BALLS dataset consisting of images of three balls bouncing in a box.

## 4.1 REGULAR AND IRREGULAR TIME GRIDS

Here we compare performance of our model on regular and irregular time grids. As Figure 8 shows, for all datasets our model performs very similarly on both types of the time grids, demonstrating its strong and robust performance on irregularly sampled data. Next, to investigate how design choices in our encoder affect the results on irregular time grids, we do an ablation study where we remove temporal attention (TA) and relative positional encodings (RPE). Note that when we remove RPE we add standard sinusoidal-cosine positional encodings as in Vaswani et al. (2017). The results are shown in Table 1. We see that removing temporal attention, or RPE, or both tends to noticeably increase test errors, indicating the effectiveness of our modifications.

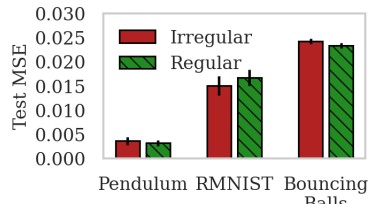

| Model | | Pendulum | RMNIST | Bouncing Balls |
|---|---|---|---|---|
| -RPE | -TA | $0.036 \pm 0.007$ | $0.068 \pm 0.000$ | $0.079 \pm 0.001$ |
| +RPE | -TA | $0.043 \pm 0.010$ | $0.062 \pm 0.002$ | $0.043 \pm 0.013$ |
| -RPE | +TA | $0.009 \pm 0.001$ | $0.047 \pm 0.002$ | $\mathbf{0.024 \pm 0.002}$ |
| +RPE | +TA | $\mathbf{0.004 \pm 0.001}$ | $\mathbf{0.015 \pm 0.002}$ | $\mathbf{0.024 \pm 0.001}$ |

Figure 8: Test errors for our model on regular and irregular time grids.

Table 1: Test MSEs for different ablations.

## 4.2 BLOCK SIZE

Our model operates on sub-trajectories whose lengths are controlled by the block sizes, i.e., the number of observations in each block (Section 3.1). Here we set the size of all blocks to a given value and demonstrate how it affects the performance of our model. Figure 9 shows test errors and training times for various block sizes. We see that the optimal block size is much smaller than the length of the observed trajectory (51 in our case), and that in some cases the model benefits from increasing the block size, but only up to some point after which the performance starts to drop. We also see how the ability to parallelize computations across block improves training times.

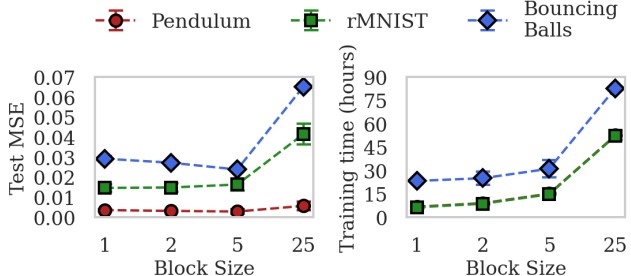

Figure 9: Test errors and training times for different block sizes.

### 4.3 CONTINUITY CONSTRAINT

Our model divides training sequences into blocks and uses the continuity prior (Equation 9) to enforce continuity of the latent trajectories across the blocks. Here we investigate how the strength of the prior (in terms of $\sigma_c$) affects the model's performance. In Figure 10 we show results for different values of $\sigma_c$. We see that stronger continuity prior tends to improve the results. For BOUNCING BALLS with $\sigma_c = 2 \cdot 10^{-5}$ the model failed to learn meaningful latent dynamics, perhaps due to excessively strong continuity prior. For new datasets the continuity prior as well as other hyperparameters can be set e.g. by cross-validation. In appendix I we also show how the value of $\sigma_c$ affects the gap between the blocks.

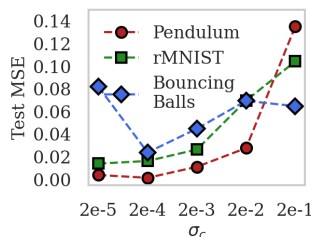

Figure 10: Test errors vs. $\sigma_c$.

### 4.4 CONSTRAINING THE APPROXIMATE POSTERIOR

We found that constraining variance of the approximate posteriors $q_{\psi_i}(s_i)$ to be at least $\tau_{\min}^2 > 0$ (in each direction) might noticeably improve performance of our model. In Figure 11 we compare the results for $\tau_{\min} = 0$ and $\tau_{\min} = 0.02$. As can be seen, this simple constraint greatly improves the model's performance on more complex datasets. This constraint could be viewed as an instance of noise injection, a technique used to improve stability of model predictions (Laskey et al., 2017; Sanchez-Gonzalez et al., 2020; Pfaff et al., 2021). Previous works inject noise into the input data, but we found that injecting noise directly in the latent space produces better results. Details are in Appendix E.4.3.

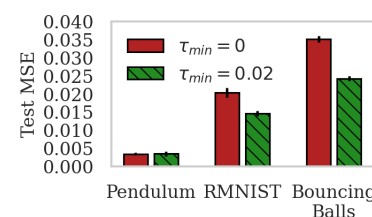

Figure 11: Errors for constrained and unconstrained approximate posteriors.

### 4.5 COMMON HEURISTICS

As discussed previously, models that compute $x_{1:N}$ directly from $x_1$ without multiple shooting (so called single shooting models) require various heuristics to train them in practice. Here we compare two commonly used heuristics with our multi-block model. First, we train our model with a single block (equivalent to single shooting) and use it as the baseline (SS). Then, we augment SS with the two heuristics and train it on short sub-trajectories (SS+sub) and on progressively increasing trajectory lengths (SS+progr). Finally, we train our sparse multiple shooting model (Ours) which is identical to SS, but has multiple blocks and continuity prior. See Appendix G for details. The results are in Figure 12. The baseline single shooting model (SS) tends to fail during training, with only a few runs converging. Hence, SS produces poor predictions on

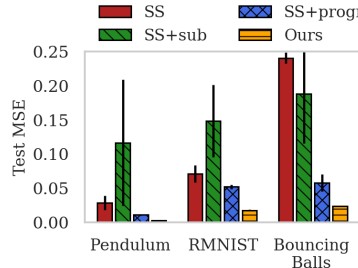

Figure 12: Errors for different heuristics.

average. Training a single shooting model on short sub-trajectories tends to make results even worse in our case. With relatively easy training, SS+sub produces unstable test predictions that quickly blow up. In our case SS+progr was the most effective heuristic, with stable training and reasonable test predictions (with a few getting a bit unstable towards the end). Compared to our model, none of the heuristics was able to match the performance of our sparse multiple shooting model.

### 4.6 COMPARISON TO OTHER MODELS

We compare our model to recent models from the literature: ODE2VAE (Yildiz et al., 2019) and NODEP (Norcliffe et al., 2021). Both models learn continuous-time deterministic dynamics in the latent space and use an encoder to map observations to the latent initial state. For comparison we use datasets on regular time grids since ODE2VAE's encoder works only on regular time grids. All models are trained and tested on full trajectories and use the first 8 observations to infer the latent initial state. We use the default parameters and code provided in the ODE2VAE and NODEP papers.

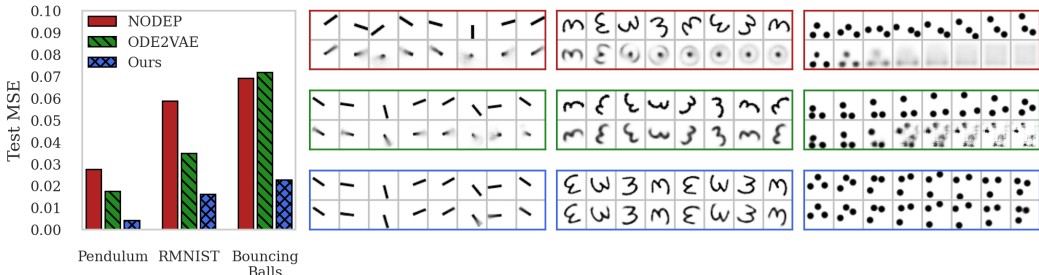

Figure 13: **Left**: Test errors for different models and datasets. **Right**: For each dataset, we plot data and predictions for NODEP, ODE2VAE and our model (top to bottom). Each sub-plot shows data as the first row, and prediction as the second row. We show prediction with the median test error. See Appendix H.4 for more predictions.

All models are trained for the same amount of time. See Appendix H for more details. Figure 13 shows the results. We see that NODEP produces reasonable predictions only for the PENDULUM dataset. ODE2VAE performs slightly better and manages to learn both PENDULUM and RMNIST data quite well, but fails on the most complex BOUNCING BALLS dataset (note that ODE2VAE uses the iterative training heuristic). Our model performs well on all three datasets. Also, see Appendix H.5 for a demonstration of the effect of the training trajectory length on NODEP and ODE2VAE.

## 5 RELATED WORK

The problem with training on long trajectories is not new and multiple shooting (MS) was proposed as a solution long time ago (van Domselaar & Hemker, 1975; Baake et al., 1992; Voss et al., 2004). Recent works have tried to adapt MS to modern neural-network-based models and large data regimes. Jordana et al. (2021) and Beintema et al. (2021) directly apply MS in latent space in fully deterministic setting, but use discrete-time dynamics without amortization or with encoders applicable only to regular time grids, and also both use ad-hoc loss terms to enforce continuity (see Appendix H.6 for comparison against our method). Hegde et al. (2022) proposed a probabilistic formulation of MS for Gaussian process based dynamics, but do not use amortization and learn dynamics directly in the data space. While not directly related to this work, recently Massaroli et al. (2021) proposed to use MS to derive a parallel-in-time ODE solver with the focus on efficient parallelization of the forward pass, but they do not explicitly consider the long trajectory problem.

Different forms of relative positional encodings (RPE) and distance-based attention were introduced in previous works, but usually for discrete and regular grids. Shaw et al. (2018) and Raffel et al. (2020) use discrete learnable RPEs which they add to keys, values or attention scores. Both works use clipping, i.e., learn RPEs only for $k$ closest points, which is some sense similar to using hardtanh function. Press et al. (2022) use discrete distance-based attention which decreases linearly with the distance. Zhao et al. (2021) use continuous learnable RPEs which are represented as an MLP which maps difference between spatial positions of two points to the corresponding RPEs which are then added to values and attention scores without clipping.

Variants of attention-based models for irregular time series were introduced in Shukla & Marlin (2021) and Zhang et al. (2020), but they are based on global positional encodings and do not constrain the size and shape of the attention windows.

## 6 CONCLUSION

In this work we developed a method that merges classical multiple shooting with principled probabilistic modeling and efficient amortized variational inference thus making the classical technique efficiently applicable in the modern large-data and large-model regimes. Our method allows to learn large-scale continuous-time dynamical systems from long observations quickly and efficiently, and, due to its probabilistic formulation, enables principled handling of noisy and partially observed data.

REPRODUCIBILITY STATEMENT

Datasets and data generation processes are described in Appendix D. Model, hyperparameters, architectures, training, validation and testing procedures, and computation algorithms are detailed in Appendices B, C, E. Source code accompanying this work will be made publicly available after review.

ACKNOWLEDGMENTS

This work was supported by NVIDIA AI Technology Center Finland.

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

## A   DEPENDENCE OF LOSS LANDSCAPE ON THE OBSERVATION INTERVAL

Here we demonstrate how complexity of the loss landscape grows with the length of the training trajectory.

For simplicity, we train a neural ODE model which is similar to the L-NODE model in Equations 1-2, but with $g_{\theta_{\text{dec}}}$ being the identity function. The dynamics function is represented by an MLP with two hidden layers of size 16 and hyperbolic tangent nonlinearities.

The training data consists of a single 2-dimensional trajectory observed over time interval of $[0, 20]$ seconds (see Figure 14). The trajectory is generated by solving the following ODE

$$\frac{d^2 x(t)}{dt^2} = -9.81 \sin{(x(t))} \tag{25}$$

with the initial position being 90 degrees (relative to the vertical) and the initial velocity being zero. The training data is generated by saving the solution of the ODE every 0.1 seconds.

We train the model with MSE loss using Adam (Kingma & Ba, 2015) optimizer and dopri5 adaptive solver from the `torchdiffeq` package (Chen et al., 2018). We start training on the first 10 points of the trajectory and double that length every 3000 iterations (hence the spikes in the loss plot in Figure 15). At the end of each 3000 iterations cycle (right before doubling the training trajectory length) we plot the loss landscape around the parameter value to which the optimizer converged. Let $\theta$ be the point to which the optimizer converged during the given cycle. We denote the corresponding loss value by a marker in Figure 15. Then, we plot the loss landscape around $\theta$ by evaluating the loss at parameter values $c\theta$, where $c \in [-4, 6]$. For the given observation time interval, the trajectory of length 10 is easy to fit, hence is considered to be "short".

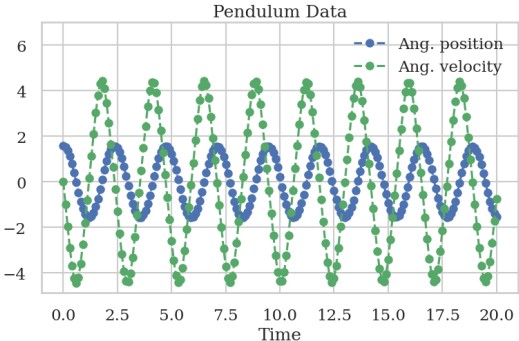

Figure 14: Pendulum data.

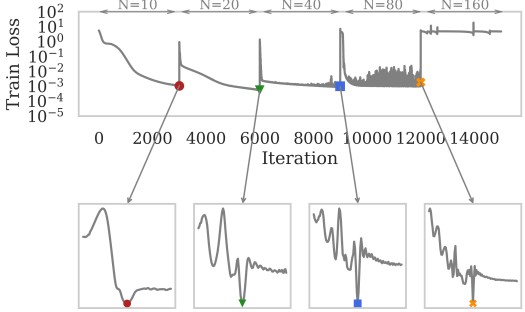

Figure 15: *Top:* Training loss of NODE model. We start with a short training trajectory ($N = 10$) and double its length at iterations denoted by the markers. Note that training fails for long enough trajectory. *Bottom:* One-dimensional projection of the loss landscape around the parameter values to which the optimizer converged for a given trajectory length. Note that complexity of the loss landscape grows with the trajectory length.

## B   MODEL, APPROXIMATE POSTERIOR, AND ELBO

Here we provide details about our model, approximate posterior and derivation of the ELBO.

**Joint distribution**   The joint distribution is

$$p(\boldsymbol{y}_{1:N}, \boldsymbol{s}_{1:B}, \theta_{\text{dyn}}, \theta_{\text{dec}}) = p(\boldsymbol{y}_{1:N}|\boldsymbol{s}_{1:B}, \theta_{\text{dyn}}, \theta_{\text{dec}})p(\boldsymbol{s}_{1:B}|\theta_{\text{dyn}})p(\theta_{\text{dyn}})p(\theta_{\text{dec}}) \tag{26}$$

with

$$p(\theta_{\text{dyn}}) = \mathcal{N}(\theta_{\text{dyn}}|\mu_{\theta_{\text{dyn}}}, \sigma^2_{\theta_{\text{dyn}}}I), \quad p(\theta_{\text{dec}}) = \mathcal{N}(\theta_{\text{dec}}|\mu_{\theta_{\text{dec}}}, \sigma^2_{\theta_{\text{dec}}}I), \tag{27}$$

$$p(\boldsymbol{s}_{1:B}|\theta_{\text{dyn}}) = p(\boldsymbol{s}_1)\prod_{b=2}^{B} p(\boldsymbol{s}_b|\boldsymbol{s}_{b-1}, \theta_{\text{dyn}}) \tag{28}$$

$$= \mathcal{N}(\boldsymbol{s}_1|\mu_0, \sigma^2_0 I)\prod_{b=2}^{B} \mathcal{N}(\boldsymbol{s}_b|\text{ODEsolve}(\boldsymbol{s}_{b-1}, t_{[b-1]}, t_{[b]}, f_{\theta_{\text{dyn}}}), \sigma^2_c I), \tag{29}$$

$$p(\boldsymbol{y}_{1:N}|\boldsymbol{s}_{1:B}, \theta_{\text{dyn}}, \theta_{\text{dec}}) = p(\boldsymbol{y}_1|\boldsymbol{s}_1, \theta_{\text{dec}})\prod_{b=1}^{B} p(\{\boldsymbol{y}_i\}_{i\in\mathcal{I}_b}|\boldsymbol{s}_b, \theta_{\text{dyn}}, \theta_{\text{dec}}) \tag{30}$$

$$= p(\boldsymbol{y}_1|\boldsymbol{s}_1, \theta_{\text{dec}})\prod_{b=1}^{B}\prod_{i\in\mathcal{I}_b} p(\boldsymbol{y}_i|\boldsymbol{s}_b, \theta_{\text{dyn}}, \theta_{\text{dec}}) \tag{31}$$

$$= \mathcal{N}(\boldsymbol{y}_1|g_{\theta_{\text{dec}}}(\boldsymbol{s}_1), \sigma^2_Y I)\prod_{b=1}^{B}\prod_{i\in\mathcal{I}_b} \mathcal{N}(\boldsymbol{y}_i|g_{\theta_{\text{dec}}}(\text{ODEsolve}(\boldsymbol{s}_b, t_{[b]}, t_i, f_{\theta_{\text{dyn}}})), \sigma^2_Y I) \tag{32}$$

$$= \mathcal{N}(\boldsymbol{y}_1|g_{\theta_{\text{dec}}}(\boldsymbol{x}_1), \sigma^2_Y I)\prod_{b=1}^{B}\prod_{i\in\mathcal{I}_b} \mathcal{N}(\boldsymbol{y}_i|g_{\theta_{\text{dec}}}(\boldsymbol{x}_i), \sigma^2_Y I), \tag{33}$$

where $\mathcal{N}$ is the Gaussian distribution, $I \in \mathbb{R}^{d\times d}$ is identity matrix, and $\sigma^2_Y$ is the observation noise variance that is shared across data dimensions.

**Approximate posterior**   The family of approximate posteriors is defined as

$$q(\theta_{\text{dyn}}, \theta_{\text{dec}}, \boldsymbol{s}_{1:B}) = q(\theta_{\text{dyn}})q(\theta_{\text{dec}})\prod_{b=1}^{B} q(\boldsymbol{s}_b) \tag{34}$$

$$= \mathcal{N}(\theta_{\text{dyn}}|\boldsymbol{\gamma}_{\theta_{\text{dyn}}}, \text{diag}(\boldsymbol{\tau}^2_{\theta_{\text{dyn}}}))\mathcal{N}(\theta_{\text{dec}}|\boldsymbol{\gamma}_{\theta_{\text{dec}}}, \text{diag}(\boldsymbol{\tau}^2_{\theta_{\text{dec}}}))\prod_{b=1}^{B} \mathcal{N}(\boldsymbol{s}_b|\boldsymbol{\gamma}_b, \text{diag}(\boldsymbol{\tau}^2_b)), \tag{35}$$

where $\text{diag}(\boldsymbol{\tau}_\bullet)$ is a matrix with vector $\boldsymbol{\tau}_\bullet$ on the main diagonal.

**ELBO**  The ELBO can be written as

$$\mathcal{L} = \int q(\theta_{\text{dyn}}, \theta_{\text{dec}}, \boldsymbol{s}_{1:B}) \ln \frac{p(\boldsymbol{y}_{1:N}, \boldsymbol{s}_{1:B}, \theta_{\text{dyn}}, \theta_{\text{dec}})}{q(\theta_{\text{dyn}}, \theta_{\text{dec}}, \boldsymbol{s}_{1:B})} d\theta_{\text{dyn}} d\theta_{\text{dec}} d\boldsymbol{s}_{1:B} \tag{36}$$

$$= \int q(\theta_{\text{dyn}}, \theta_{\text{dec}}, \boldsymbol{s}_{1:B}) \ln \frac{p(\boldsymbol{y}_{1:N}|\boldsymbol{s}_{1:B}, \theta_{\text{dyn}}, \theta_{\text{dec}}) p(\boldsymbol{s}_{1:B}|\theta_{\text{dyn}}) p(\theta_{\text{dyn}}) p(\theta_{\text{dec}})}{q(\boldsymbol{s}_{1:B}) q(\theta_{\text{dyn}}) q(\theta_{\text{dec}})} d\theta_{\text{dyn}} d\theta_{\text{dec}} d\boldsymbol{s}_{1:B} \tag{37}$$

$$= \int q(\theta_{\text{dyn}}, \theta_{\text{dec}}, \boldsymbol{s}_{1:B}) \ln p(\boldsymbol{y}_{1:N}|\boldsymbol{s}_{1:B}, \theta_{\text{dyn}}, \theta_{\text{dec}}) d\theta_{\text{dyn}} d\theta_{\text{dec}} d\boldsymbol{s}_{1:B} \tag{38}$$

$$- \int q(\theta_{\text{dyn}}, \theta_{\text{dec}}, \boldsymbol{s}_{1:B}) \ln \frac{q(\boldsymbol{s}_{1:B})}{p(\boldsymbol{s}_{1:B}|\theta_{\text{dyn}})} d\theta_{\text{dyn}} d\theta_{\text{dec}} d\boldsymbol{s}_{1:B} \tag{39}$$

$$- \int q(\theta_{\text{dyn}}, \theta_{\text{dec}}, \boldsymbol{s}_{1:B}) \ln \frac{q(\theta_{\text{dyn}})}{p(\theta_{\text{dyn}})} d\theta_{\text{dyn}} d\theta_{\text{dec}} d\boldsymbol{s}_{1:B} \tag{40}$$

$$- \int q(\theta_{\text{dec}}, \theta_{\text{dec}}, \boldsymbol{s}_{1:B}) \ln \frac{q(\theta_{\text{dec}})}{p(\theta_{\text{dec}})} d\theta_{\text{dyn}} d\theta_{\text{dec}} d\boldsymbol{s}_{1:B} \tag{41}$$

$$= \mathcal{L}_1 - \mathcal{L}_2 - \mathcal{L}_3 - \mathcal{L}_4. \tag{42}$$

Let's look at each term $\mathcal{L}_i$ separately.

$$\mathcal{L}_1 = \int q(\theta_{\text{dyn}}, \theta_{\text{dec}}, \boldsymbol{s}_{1:B}) \ln p(\boldsymbol{y}_{1:N}|\boldsymbol{s}_{1:B}, \theta_{\text{dyn}}, \theta_{\text{dec}}) d\theta_{\text{dyn}} d\theta_{\text{dec}} d\boldsymbol{s}_{1:B} \tag{43}$$

$$= \int q(\theta_{\text{dyn}}, \theta_{\text{dec}}, \boldsymbol{s}_{1:B}) \ln \left[ p(\boldsymbol{y}_1|\boldsymbol{s}_1, \theta_{\text{dec}}) \prod_{b=1}^{B} p(\{\boldsymbol{y}_i\}_{i \in \mathcal{I}_b}|\boldsymbol{s}_b, \theta_{\text{dyn}}, \theta_{\text{dec}}) \right] d\theta_{\text{dyn}} d\theta_{\text{dec}} d\boldsymbol{s}_{1:B} \tag{44}$$

$$= \int q(\theta_{\text{dyn}}, \theta_{\text{dec}}, \boldsymbol{s}_{1:B}) \ln p(\boldsymbol{y}_1|\boldsymbol{s}_1, \theta_{\text{dec}}) d\theta_{\text{dyn}} d\theta_{\text{dec}} d\boldsymbol{s}_{1:B} \tag{45}$$

$$+ \int q(\theta_{\text{dyn}}, \theta_{\text{dec}}, \boldsymbol{s}_{1:B}) \ln \left[ \prod_{b=1}^{B} p(\{\boldsymbol{y}_i\}_{i \in \mathcal{I}_b}|\boldsymbol{s}_b, \theta_{\text{dyn}}, \theta_{\text{dec}}) \right] d\theta_{\text{dyn}} d\theta_{\text{dec}} d\boldsymbol{s}_{1:B} \tag{46}$$

$$= \int q(\theta_{\text{dyn}}, \theta_{\text{dec}}, \boldsymbol{s}_{1:B}) \ln p(\boldsymbol{y}_1|\boldsymbol{s}_1, \theta_{\text{dec}}) d\theta_{\text{dyn}} d\theta_{\text{dec}} d\boldsymbol{s}_{1:B} \tag{47}$$

$$+ \sum_{b=1}^{B} \int q(\theta_{\text{dyn}}, \theta_{\text{dec}}, \boldsymbol{s}_{1:B}) \ln p(\{\boldsymbol{y}_i\}_{i \in \mathcal{I}_b}|\boldsymbol{s}_b, \theta_{\text{dyn}}, \theta_{\text{dec}}) d\theta_{\text{dyn}} d\theta_{\text{dec}} d\boldsymbol{s}_{1:B} \tag{48}$$

$$= \int q(\theta_{\text{dec}}, \boldsymbol{s}_1) \ln p(\boldsymbol{y}_1|\boldsymbol{s}_1, \theta_{\text{dec}}) d\theta_{\text{dec}} d\boldsymbol{s}_1 \tag{49}$$

$$+ \sum_{b=1}^{B} \int q(\theta_{\text{dyn}}, \theta_{\text{dec}}, \boldsymbol{s}_b) \ln p(\{\boldsymbol{y}_i\}_{i \in \mathcal{I}_b}|\boldsymbol{s}_b, \theta_{\text{dyn}}, \theta_{\text{dec}}) d\theta_{\text{dyn}} d\theta_{\text{dec}} d\boldsymbol{s}_b \tag{50}$$

$$= \mathbb{E}_{q(\theta_{\text{dec}}, \boldsymbol{s}_1)} \left[ \ln p(\boldsymbol{y}_1|\boldsymbol{s}_1, \theta_{\text{dec}}) \right] + \sum_{b=1}^{B} \mathbb{E}_{q(\theta_{\text{dyn}}, \theta_{\text{dec}}, \boldsymbol{s}_b)} \left[ \ln p(\{\boldsymbol{y}_i\}_{i \in \mathcal{I}_b}|\boldsymbol{s}_b, \theta_{\text{dyn}}, \theta_{\text{dec}}) \right] \tag{51}$$

$$= \mathbb{E}_{q(\theta_{\text{dec}}, \boldsymbol{s}_1)} \left[ \ln p(\boldsymbol{y}_1|\boldsymbol{s}_1, \theta_{\text{dec}}) \right] + \sum_{b=1}^{B} \sum_{i \in \mathcal{I}_b} \mathbb{E}_{q(\theta_{\text{dyn}}, \theta_{\text{dec}}, \boldsymbol{s}_b)} \left[ \ln p(\boldsymbol{y}_i|\boldsymbol{s}_b, \theta_{\text{dyn}}, \theta_{\text{dec}}) \right] \tag{52}$$

$$\mathcal{L}_2 = \int q(\theta_{\text{dyn}}, \theta_{\text{dec}}, \boldsymbol{s}_{1:B}) \ln \frac{q(\boldsymbol{s}_{1:B})}{p(\boldsymbol{s}_{1:B}|\theta_{\text{dyn}})} d\theta_{\text{dyn}} d\theta_{\text{dec}} d\boldsymbol{s}_{1:B} \tag{53}$$

$$= \int q(\theta_{\text{dyn}}, \theta_{\text{dec}}, \boldsymbol{s}_{1:B}) \ln \left[ \frac{q(\boldsymbol{s}_1)}{p(\boldsymbol{s}_1)} \prod_{b=2}^{B} \frac{q(\boldsymbol{s}_b)}{p(\boldsymbol{s}_b|\boldsymbol{s}_{b-1}, \theta_{\text{dyn}})} \right] d\theta_{\text{dyn}} d\theta_{\text{dec}} d\boldsymbol{s}_{1:B} \tag{54}$$

$$= \int q(\theta_{\text{dyn}}, \theta_{\text{dec}}, \boldsymbol{s}_{1:B}) \ln \left[ \frac{q(\boldsymbol{s}_1)}{p(\boldsymbol{s}_1)} \right] d\theta_{\text{dyn}} d\theta_{\text{dec}} d\boldsymbol{s}_{1:B} \tag{55}$$

$$+ \int q(\theta_{\text{dyn}}, \theta_{\text{dec}}, \boldsymbol{s}_{1:B}) \ln \left[ \prod_{b=2}^{B} \frac{q(\boldsymbol{s}_b)}{p(\boldsymbol{s}_b|\boldsymbol{s}_{b-1}, \theta_{\text{dyn}})} \right] d\theta_{\text{dyn}} d\theta_{\text{dec}} d\boldsymbol{s}_{1:B} \tag{56}$$

$$= \int q(\theta_{\text{dyn}}, \theta_{\text{dec}}, \boldsymbol{s}_{1:B}) \ln \left[ \frac{q(\boldsymbol{s}_1)}{p(\boldsymbol{s}_1)} \right] d\theta_{\text{dyn}} d\theta_{\text{dec}} d\boldsymbol{s}_{1:B} \tag{57}$$

$$+ \sum_{b=2}^{B} \int q(\theta_{\text{dyn}}, \theta_{\text{dec}}, \boldsymbol{s}_{1:B}) \ln \left[ \frac{q(\boldsymbol{s}_b)}{p(\boldsymbol{s}_b|\boldsymbol{s}_{b-1}, \theta_{\text{dyn}})} \right] d\theta_{\text{dyn}} d\theta_{\text{dec}} d\boldsymbol{s}_{1:B} \tag{58}$$

$$= \int q(\boldsymbol{s}_1) \ln \left[ \frac{q(\boldsymbol{s}_1)}{p(\boldsymbol{s}_1)} \right] d\boldsymbol{s}_1 \tag{59}$$

$$+ \sum_{b=2}^{B} \int q(\theta_{\text{dyn}}, \boldsymbol{s}_{b-1}, \boldsymbol{s}_b) \ln \left[ \frac{q(\boldsymbol{s}_b)}{p(\boldsymbol{s}_b|\boldsymbol{s}_{b-1}, \theta_{\text{dyn}})} \right] d\theta_{\text{dyn}} d\boldsymbol{s}_{b-1} d\boldsymbol{s}_b \tag{60}$$

$$= \int q(\boldsymbol{s}_1) \ln \left[ \frac{q(\boldsymbol{s}_1)}{p(\boldsymbol{s}_1)} \right] d\boldsymbol{s}_1 \tag{61}$$

$$+ \sum_{b=2}^{B} \int q(\theta_{\text{dyn}}, \boldsymbol{s}_{b-1}) \left( \int q(\boldsymbol{s}_b) \ln \left[ \frac{q(\boldsymbol{s}_b)}{p(\boldsymbol{s}_b|\boldsymbol{s}_{b-1}, \theta_{\text{dyn}})} \right] d\boldsymbol{s}_b \right) d\theta_{\text{dyn}} d\boldsymbol{s}_{b-1} \tag{62}$$

$$= \int q(\boldsymbol{s}_1) \ln \left[ \frac{q(\boldsymbol{s}_1)}{p(\boldsymbol{s}_1)} \right] d\boldsymbol{s}_1 \tag{63}$$

$$+ \sum_{b=2}^{B} \int q(\theta_{\text{dyn}}, \boldsymbol{s}_{b-1}) \text{KL}\left( q(\boldsymbol{s}_b) \| p(\boldsymbol{s}_b|\boldsymbol{s}_{b-1}, \theta_{\text{dyn}}) \right) d\theta_{\text{dyn}} d\boldsymbol{s}_{b-1} \tag{64}$$

$$= \text{KL}\left( q(\boldsymbol{s}_1) \| p(\boldsymbol{s}_1) \right) + \sum_{b=2}^{B} \mathbb{E}_{q(\theta_{\text{dyn}}, \boldsymbol{s}_{b-1})} \left[ \text{KL}\left( q(\boldsymbol{s}_b) \| p(\boldsymbol{s}_b|\boldsymbol{s}_{b-1}, \theta_{\text{dyn}}) \right) \right], \tag{65}$$

where KL is Kullback–Leibler divergence.

$$\mathcal{L}_3 = \text{KL}(q(\theta_{\text{dyn}}) \| p(\theta_{\text{dyn}})), \quad \mathcal{L}_4 = \text{KL}(q(\theta_{\text{dec}}) \| p(\theta_{\text{dec}})). \tag{66}$$

**Computing ELBO** All expectations are approximated using Monte Carlo integration with one sample, that is

$$\mathbb{E}_{p(z)}[f(z)] \approx f(\zeta), \quad \text{where } \zeta \text{ is sampled from } p(z). \tag{67}$$

The KL terms contain only Gaussian distributions, so can be computed in closed form.

## C    COMPUTATION ALGORITHMS

### C.1    ELBO

To find the approximate posterior which minimizes the Kullback–Leibler divergence

$$\mathrm{KL}(q(\theta_{\mathrm{dyn}}, \theta_{\mathrm{dec}}, \boldsymbol{s}_{1:B}) \| p(\theta_{\mathrm{dyn}}, \theta_{\mathrm{dec}}, \boldsymbol{s}_{1:B} | \boldsymbol{y}_{1:N})), \tag{68}$$

we maximize the evidence lower bound (ELBO) which for our model is defined as

$$\mathcal{L} = \underbrace{\mathbb{E}_{q(\theta_{\mathrm{dec}}, \boldsymbol{s}_1)} \big[ \log p(\boldsymbol{y}_1 | \boldsymbol{s}_1, \theta_{\mathrm{dec}}) \big]}_{\textit{(i) data likelihood}} + \sum_{b=1}^{B} \sum_{i \in \mathcal{I}_b} \underbrace{\mathbb{E}_{q(\theta_{\mathrm{dyn}}, \theta_{\mathrm{dec}}, \boldsymbol{s}_b)} \big[ \log p(\boldsymbol{y}_i | \boldsymbol{s}_b, \theta_{\mathrm{dyn}}, \theta_{\mathrm{dec}}) \big]}_{\textit{(ii) data likelihood}} \tag{69}$$

$$- \underbrace{\mathrm{KL}\big[q(\boldsymbol{s}_1) \| p(\boldsymbol{s}_1)\big]}_{\textit{(iii) initial state prior}} - \sum_{b=2}^{B} \mathbb{E}_{q(\theta_{\mathrm{dyn}}, \boldsymbol{s}_{b-1})} \Big[ \underbrace{\mathrm{KL}\big[q(\boldsymbol{s}_b) \| p(\boldsymbol{s}_b | \boldsymbol{s}_{b-1}, \theta_{\mathrm{dyn}})\big]}_{\textit{(iv) continuity prior}} \Big] \tag{70}$$

$$- \underbrace{\mathrm{KL}\big[q(\theta_{\mathrm{dyn}}) \| p(\theta_{\mathrm{dyn}})\big]}_{\textit{(v) dynamics prior}} - \underbrace{\mathrm{KL}\big[q(\theta_{\mathrm{dec}}) \| p(\theta_{\mathrm{dec}})\big]}_{\textit{(vi) decoder prior}}. \tag{71}$$

The ELBO is computed using the following algorithm:

1. Sample $\theta_{\mathrm{dyn}}, \theta_{\mathrm{dec}}$ from $q_{\boldsymbol{\psi}_{\mathrm{dyn}}}(\theta_{\mathrm{dyn}}), q_{\boldsymbol{\psi}_{\mathrm{dec}}}(\theta_{\mathrm{dec}})$.
2. Sample $\boldsymbol{s}_{1:B}$ from $q_{\boldsymbol{\psi}_1}(\boldsymbol{s}_1), ..., q_{\boldsymbol{\psi}_B}(\boldsymbol{s}_B)$ with $\boldsymbol{\psi}_{1:B} = h_{\theta_{\mathrm{enc}}}(\boldsymbol{y}_{1:N})$.
3. Compute $\boldsymbol{x}_{1:N}$ from $\boldsymbol{s}_{1:B}$ as in Equations 11-12.
4. Compute ELBO $\mathcal{L}$ (KL terms are computed in closed form, for expectations we use Monte Carlo integration with one sample).

Sampling is done using reparametrization to allow unbiased gradients w.r.t. the model parameters.

We observed that under some hyper-parameter configurations the continuity-promoting term *(iv)* might cause the shooting variables to collapse to a single point hence preventing the learning of meaningful dynamics. Downscaling this term helps to avoid the collapse. However, in our experiments we did not use any scaling.

### C.2    FORECASTING

Given initial observations $\boldsymbol{y}_{1:N_1}^*$ at time points $t_{1:N_1}^*$ we predict the future observations $\boldsymbol{y}_{N_1+1:N_2}^*$ at time points $t_{N_1+1:N_2}^*$ as the expected value of the (approximate) posterior predictive distribution

$$p(\boldsymbol{y}_{N_1+1:N_2}^* | \boldsymbol{y}_{1:N_1}^*, \boldsymbol{y}_{1:N}) \approx \int p(\boldsymbol{y}_{N_1+1:N_2}^* | \boldsymbol{s}_1^*, \theta_{\mathrm{dyn}}, \theta_{\mathrm{dec}}) q_{\boldsymbol{\psi}_1^*}(\boldsymbol{s}_1^*) q_{\boldsymbol{\psi}_{\mathrm{dyn}}}(\theta_{\mathrm{dyn}}) q_{\boldsymbol{\psi}_{\mathrm{dec}}}(\theta_{\mathrm{dec}}) d\boldsymbol{s}_1^* d\theta_{\mathrm{dyn}} d\theta_{\mathrm{dec}}, \tag{72}$$

where $\boldsymbol{\psi}_1^* = h_{\theta_{\mathrm{enc}}}(\boldsymbol{y}_{1:N_1}^*)$. The expected value is estimated via Monte Carlo integration, so the algorithm for predicting $\boldsymbol{y}_{N_1+1:N_2}^*$ is

1. Sample $\theta_{\mathrm{dyn}}, \theta_{\mathrm{dec}}$ from $q_{\boldsymbol{\psi}_{\mathrm{dyn}}}(\theta_{\mathrm{dyn}}), q_{\boldsymbol{\psi}_{\mathrm{dec}}}(\theta_{\mathrm{dec}})$.
2. Sample $\boldsymbol{s}_1^*$ from $q_{\boldsymbol{\psi}_1^*}(\boldsymbol{s}_1^*)$ with $\boldsymbol{\psi}_1^* = h_{\theta_{\mathrm{enc}}}(\boldsymbol{y}_{1:N_1}^*)$.
3. Calculate latent states $\boldsymbol{x}_i = \mathrm{ODEsolve}(\boldsymbol{s}_1^*, t_1^*, t_i^*, f_{\theta_{\mathrm{dyn}}}), \quad i \in \{N_1 + 1, ..., N_2\}$.
4. Sample $\boldsymbol{y}_i^*$ from $p(\boldsymbol{y}_i^* | g_{\theta_{\mathrm{dec}}}(\boldsymbol{x}_i)), \quad i \in \{N_1 + 1, ..., N_2\}$.
5. Repeat steps 1-4 $n$ times and average the predicted trajectories $\boldsymbol{y}_{N_1+1:N_2}^*$ (we use $n = 10$).

# D   DATASETS

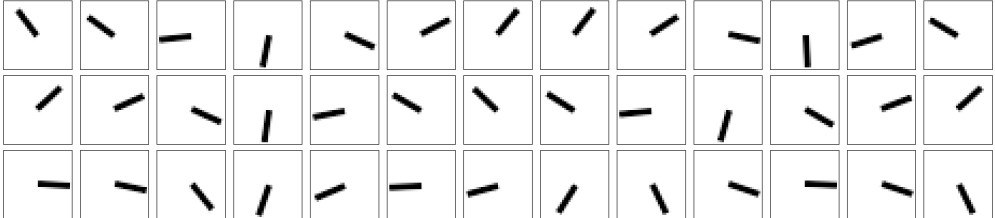

Figure 16: Examples of trajectories from the PENDULUM dataset.

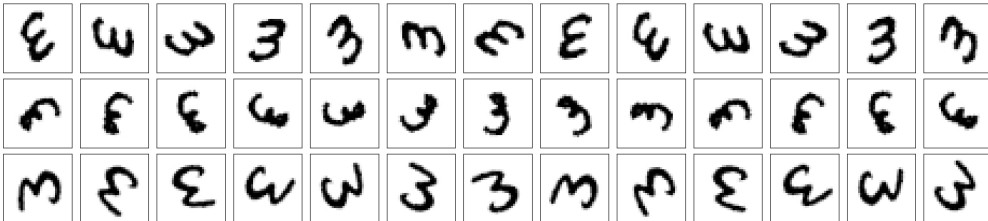

Figure 17: Examples of trajectories from the RMNIST dataset.

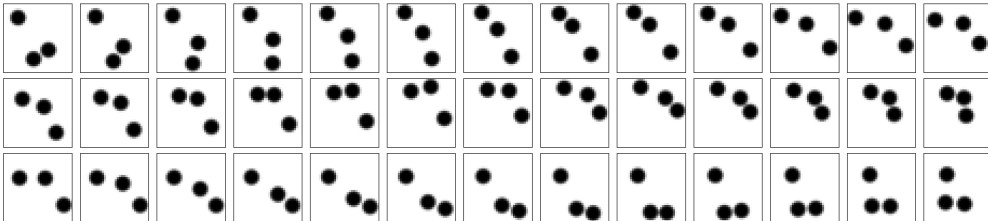

Figure 18: Examples of trajectories from the BOUNCING BALLS dataset.

Here we provide details about the datasets used in this work and about the data generation procedures. The datasets we selected are commonly used in literature concerned with modeling of temporal processes (Karl et al., 2017; Ha et al., 2019; Casale et al., 2018; Yildiz et al., 2019; Norcliffe et al., 2021; Sutskever et al., 2008; Lotter et al., 2015; Hsieh et al., 2018; Gan et al., 2015). To the best of our knowledge, previous works consider these datasets only on regular time grids (i.e., the temporal distance between consecutive observations is constant). Since in this work we are mostly interested in processes observed at irregular time intervals, we generate these datasets on both regular and irregular time grids. The datasets and data generation scripts can be downloaded at https://github.com/yakovlev31/msvi.

## D.1   PENDULUM

This dataset consist of images of a pendulum moving under the influence of gravity. Each trajectory is generated by sampling the initial angle $x$ and angular velocity $\dot{x}$ of the pendulum and simulating its dynamics over a period of time. The algorithm for simulating one trajectory is

1. Sample $x \sim \text{Uniform}[0, 2\pi]$ (in rads) and $\dot{x} \sim \text{Uniform}[-\pi/2, \pi/2]$ (in rads/second).
2. Generate time grid $(t_1, ..., t_N)$. Regular time grids are generated by placing the time points at equal distances along the time interval $[t_1, t_N]$ with the first time point placed at $t_1$ and the last time point placed at $t_N$. Irregular time grids are generated by sampling $N$ points from the time interval $[t_1, t_N]$ uniformly at random with the first time point placed at $t_1$, the last time point placed at $t_N$, and also ensuring that the minimum distance between time points is larger than $\frac{t_N - t_1}{4(N-1)}$ (i.e., a quarter of the time step of a regular time grid).

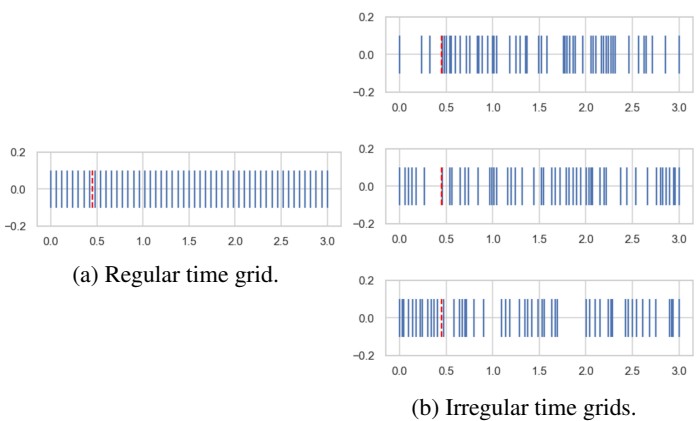

(a) Regular time grid.

(b) Irregular time grids.

Figure 19: Examples of regular and irregular time grids for PENDULUM dataset. At test time, observations before the red lines are used to compute the latent initial state.

3. Solve the ODE $\frac{d^2\boldsymbol{x}(t)}{dt^2} = -9.81\sin(\boldsymbol{x}(t))$ with initial state $\boldsymbol{x}, \dot{\boldsymbol{x}}$ at time points $(t_1, ..., t_N)$.

4. Create sequence of observations $(\boldsymbol{y}_1, ..., \boldsymbol{y}_N)$ with $\boldsymbol{y}_i = \text{observe}(\boldsymbol{x}(t_i))$, where $\boldsymbol{x}(t_i)$ is the solution of the ODE above at time point $t_i$ and $\text{observe}(\cdot)$ is a mapping from the pendulum angle to the corresponding observation.

The training/validation/test sets contain 400/50/50 trajectories. Regular time grids are identical across all trajectories. Irregular time grids are unique for each trajectory. The only constraint we place on the time grids is that they contain $N$ time points (for efficient implementation and meaningful comparison). We set $t_1 = 0$, $t_N = 3$, and $N = 51$. Each observation $\boldsymbol{y}_i$ is a 1024-dimensional vector (flat $32 \times 32$ image).

### D.2 RMNIST

This dataset consist of images of rotating digits 3 sampled from the MNIST dataset. Each trajectory is generated by sampling a digit 3 from the MNIST dataset uniformly at random without replacement, then sampling the initial angle $\boldsymbol{x}$ and angular velocity $\dot{\boldsymbol{x}}$ and simulating the frictionless rotation of the digit. The algorithm for simulating one trajectory is

1. Sample a digit 3 from the MNIST dataset uniformly at random without replacement.

2. Sample $\boldsymbol{x} \sim \text{Uniform}[0, 2\pi]$ (in rads) and $\dot{\boldsymbol{x}} \sim \text{Uniform}[\pi, 2\pi]$ (in rads/second).

3. Generate time grid $(t_1, ..., t_N)$. Regular time grids are generated by placing the time points at equal distances along the time interval $[t_1, t_N]$ with the first time point placed at $t_1$ and the last time point placed at $t_N$. Irregular time grids are generated by sampling $N$ points from the time interval $[t_1, t_N]$ uniformly at random with the first time point placed at $t_1$, the last time point placed at $t_N$, and also ensuring that the minimum distance between time points is larger than $\frac{t_N - t_1}{4(N-1)}$ (i.e., a quarter of the time step of a regular time grid).

4. Solve the ODE $\frac{d\boldsymbol{x}(t)}{dt} = \dot{\boldsymbol{x}}$ with initial state $\boldsymbol{x}$ at time points $(t_1, ..., t_N)$.

5. Create sequence of observations $(\boldsymbol{y}_1, ..., \boldsymbol{y}_N)$ with $\boldsymbol{y}_i = \text{observe}(\boldsymbol{x}(t_i))$, where $\boldsymbol{x}(t_i)$ is the solution of the ODE above at time point $t_i$ and $\text{observe}(\cdot)$ is a mapping from the digit angle to the corresponding observation.

The training/validation/test sets contain 4000/500/500 trajectories. Regular time grids are identical across all trajectories. Irregular time grids are unique for each trajectory. The only constraint we place on the time grids is that they contain $N$ time points (for efficient implementation and meaningful comparison). We set $t_1 = 0$, $t_N = 2$, and $N = 51$. Each observation $\boldsymbol{y}_i$ is a 1024-dimensional vector (flat $32 \times 32$ image).

### D.3 BOUNCING BALLS

This dataset consist of images of three balls bouncing in a frictionless box. Each trajectory is generated by sampling the initial positions and velocities of the three balls and simulating the frictionless collision dynamics. The algorithm for simulating one trajectory is

1. Sample initial positions of the three balls uniformly at random such that the balls do not overlap and do not extend outside the boundaries of the box.

2. Sample initial velocities of the three balls $v \in \mathbb{R}^3$ as $v = \frac{v'}{\|v'\|}$, where $v'$ is sampled from the standard normal distribution.

3. Generate time grid $(t_1, ..., t_N)$. Regular time grids are generated by placing the time points at equal distances along the time interval $[t_1, t_N]$ with the first time point placed at $t_1$ and the last time point placed at $t_N$. Irregular time grids are generated by sampling $N$ points from the time interval $[t_1, t_N]$ uniformly at random with the first time point placed at $t_1$, the last time point placed at $t_N$, and also ensuring that the minimum distance between time points is larger than $\frac{t_N - t_1}{4(N-1)}$ (i.e., a quarter of the time step of a regular time grid).

4. Solve the ODE representing the frictionless collision dynamics at time points $(t_1, ..., t_N)$ (see the data generating script for details).

5. Create sequence of observations $(y_1, ..., y_N)$ with $y_i = \text{observe}(\theta(t_i))$, where $\theta(t_i)$ is the solution of the ODE above at time point $t_i$ and $\text{observe}(...)$ is a mapping from positions of the balls to the corresponding observation.

The training/validation/test sets contain 10000/1000/1000 trajectories. Regular time grids are identical across all trajectories. Irregular time grids are unique for each trajectory. The only constraint we place on the time grids is that they contain $N$ time points (for efficient implementation and meaningful comparison). We set $t_1 = 0$, $t_N = 20$, and $N = 51$. Each observation $y_i$ is a 1024-dimensional vector (flat $32 \times 32$ image).

## E SETUP

### E.1 TRAINING, VALIDATION, TESTING

#### E.1.1 DATA PREPROCESSING

We normalize the observations by the maximum absolute value in the training set.

#### E.1.2 TRAINING

We train our model for 300000 iterations using Adam optimizer (Kingma & Ba, 2015) with learning rate exponentially decreasing from 3e-4 to 1e-5. To simulate the model's dynamics we use differentiable ODE solvers from `torchdiffeq` package (Chen et al., 2018). In particular, we use the dopri5 solver with rtol = atol = $10^{-5}$ without the adjoint method. For PENDULUM, RMNIST, and BOUNCING BALLS datasets the batch size is set to 16, 16, and 64, respectively, while the block size is set to 1, 1, and 5, respectively. For some datasets we use data augmentation: PENDULUM - horizontal flip, BOUNCING BALLS - vertical and horizontal flips. For each dataset we set $\delta_r$ to 15% of the corresponding observation interval $[t_1, t_N]$.

#### E.1.3 VALIDATION

We use validation set to track performance of the model during training and save the parameters that produce the best validation performance. As performance measure we use the mean squared error at predicting the full validation trajectories given some number of initial observations. We use all observations within the interval $[t_1, t_1 + \delta_{\text{test}}]$ as initial observations from which we infer the latent initial state. As during training, we set $\delta_{\text{test}}$ to 15% of the observation interval $[t_1, t_N]$. The predictions are made as described in Section 3.2 but with a single sample from the posterior.

### E.1.4 TESTING

Predictions for the test trajectories are made as described in Section 3.2. Similarly to validation, we use all observations within the interval $[t_1, t_1 + \delta_{\text{test}}]$ as initial observations from which we predict the latent initial state. We set $\delta_{\text{test}}$ to 15% of the observation interval $[t_1, t_N]$.

### E.2 PRIORS

As discussed in Appendix B, we use the following priors:

$$p(\theta_{\text{dyn}}) = \mathcal{N}(\theta_{\text{dyn}}|\mu_{\theta_{\text{dyn}}}, \sigma^2_{\theta_{\text{dyn}}}I), \quad p(\theta_{\text{dec}}) = \mathcal{N}(\theta_{\text{dec}}|\mu_{\theta_{\text{dec}}}, \sigma^2_{\theta_{\text{dec}}}I), \tag{73}$$

$$p(\boldsymbol{s}_{1:B}|\theta_{\text{dyn}}) = \mathcal{N}(\boldsymbol{s}_1|\mu_0, \sigma_0^2 I)\prod_{b=2}^{B}\mathcal{N}(\boldsymbol{s}_b|\text{ODEsolve}(\boldsymbol{s}_{b-1}, t_{[b-1]}, t_{[b]}, f_{\theta_{\text{dyn}}}), \sigma_c^2 I). \tag{74}$$

We set $\mu_{\theta_{\text{dyn}}} = \mu_{\theta_{\text{dec}}} = \boldsymbol{0}$, $\sigma_{\theta_{\text{dyn}}} = \sigma_{\theta_{\text{dec}}} = 1$, $\mu_0 = \boldsymbol{0}$, $\sigma_0 = 1$, and $\sigma_c = \frac{\xi}{\sqrt{d}}$, where $\xi$ denotes the required average distance between $\boldsymbol{s}_i$ and $\boldsymbol{x}_i$, and $d$ is the latent space dimension. In this work we use $d = 32$. The parameter $\xi$ is dataset specific, for PENDULUM and RMNIST we set $\xi = 10^{-4}$, for BOUNCING BALLS we set $\xi = 10^{-3}$.

### E.3 VARIATIONAL PARAMETERS

As discussed in Appendix B, we use the following family of approximate posteriors:

$$q(\theta_{\text{dyn}}, \theta_{\text{dec}}, \boldsymbol{s}_{1:B}) = \mathcal{N}(\theta_{\text{dyn}}|\boldsymbol{\gamma}_{\theta_{\text{dyn}}}, \text{diag}(\boldsymbol{\tau}^2_{\theta_{\text{dyn}}}))\mathcal{N}(\theta_{\text{dec}}|\boldsymbol{\gamma}_{\theta_{\text{dec}}}, \text{diag}(\boldsymbol{\tau}^2_{\theta_{\text{dec}}}))\prod_{b=1}^{B}\mathcal{N}(\boldsymbol{s}_b|\boldsymbol{\gamma}_b, \text{diag}(\boldsymbol{\tau}^2_b)) \tag{75}$$

While $\boldsymbol{\gamma}_b$ and $\boldsymbol{\tau}_b$ are provided by the encoder, other variational parameters are directly optimized. We initialize $\boldsymbol{\gamma}_{\theta_{\text{dyn}}}$ and $\boldsymbol{\gamma}_{\theta_{\text{dec}}}$ using default Xavier (Glorot & Bengio, 2010) initialization of the dynamics function and decoder (see PyTorch 1.12 (Paszke et al., 2019) documentation for details). We initialize $\boldsymbol{\tau}_{\theta_{\text{dyn}}}$ and $\boldsymbol{\tau}_{\theta_{\text{dec}}}$ as vectors with each element equal to $9 \cdot 10^{-4}$.

### E.4 MODEL ARCHITECTURE

### E.4.1 DYNAMICS FUNCTION

Many physical systems, including the ones we consider in this work, are naturally modeled using second order dynamics. We structure the latent space and dynamics function so that we include this useful inductive bias into our model. In particular, we follow Yildiz et al. (2019) and split the latent space into two parts representing "position" and "velocity". That is, we represent the latent state $\boldsymbol{x}(t) \in \mathbb{R}^d$ as a concatenation of two components:

$$\boldsymbol{x}(t) = \begin{pmatrix} \boldsymbol{x}_{\text{p}}(t) \\ \boldsymbol{x}_{\text{v}}(t) \end{pmatrix}, \tag{76}$$

where $\boldsymbol{x}_{\text{p}}(t) \in \mathbb{R}^{d/2}$ is the position component and $\boldsymbol{x}_{\text{v}}(t) \in \mathbb{R}^{d/2}$ is the velocity component.

Then, we represent the dynamics function $f_{\theta_{\text{dyn}}}(t, \boldsymbol{x}(t))$ as

$$f_{\theta_{\text{dyn}}}(t, \boldsymbol{x}(t)) = \begin{pmatrix} \boldsymbol{x}_{\text{v}}(t) \\ f^{\text{v}}_{\theta_{\text{dyn}}}(t, \boldsymbol{x}(t)) \end{pmatrix}, \tag{77}$$

where $f^{\text{v}}_{\theta_{\text{dyn}}}(t, \boldsymbol{x}(t)) : \mathbb{R} \times \mathbb{R}^d \to \mathbb{R}^{d/2}$ is the dynamics function modeling the instantaneous rate of change of the velocity component.

In all our experiments we remove the dependence of $f^{\text{v}}_{\theta_{\text{dyn}}}$ on time $t$ and represent it as a multi-layer perceptron whose architecture depends on the dataset:

- PENDULUM: input size $d$, output size $d/2$, two hidden layers with size 256 and ReLU nonlinearities.

- RMNIST: input size $d$, output size $d/2$, two hidden layers with size 512 and ReLU nonlinearities.
- BOUNCING BALLS: input size $d$, output size $d/2$, three hidden layers with size 1024 and ReLU nonlinearities.

In this work we use $d = 32$.

### E.4.2 DECODER

The decoder $g_{\theta_{\text{dec}}}$ maps the latent state $\boldsymbol{x}_i$ to parameters of $p(\boldsymbol{y}_i|g_{\theta_{\text{dec}}}(\boldsymbol{x}_i))$. As we discussed in Appendix B, we set $p(\boldsymbol{y}_i|g_{\theta_{\text{dec}}}(\boldsymbol{x}_i)) = \mathcal{N}(\boldsymbol{y}_i|g_{\theta_{\text{dec}}}(\boldsymbol{x}_i), \sigma_Y^2 I)$, so the decoder outputs the mean of a Gaussian distribution. We treat $\sigma_Y$ as a hyperparameter and set it to $10^{-3}$. In our experiments, trying to learn $\sigma_Y$ resulted in overfitting. Following Yildiz et al. (2019), our encoder utilizes only the "position" part $\boldsymbol{x}_i^{\text{p}}$ of the latent state $\boldsymbol{x}_i$ since this part is assumed to contain all the information required to reconstruct the observations (see Appendix E.4.1).

We represent $g_{\theta_{\text{dec}}}$ as the composition of a convolutional neural network (CNN) with a sigmoid function to keep the mean in the interval $(0, 1)$. In particular, $g_{\theta_{\text{dec}}}$ has the following architecture: linear layer, four transposed convolution layers (2x2 kernel, stride 2) with batch norm and ReLU nonlinearities, convolutional layer (5x5 kernel, padding 2), sigmoid function. The four transposed convolution layers have $8n$, $4n$, $2n$ and $n$ channels, respectively. The convolution layer has $n$ channels. For datasets PENDULUM, RMNIST, and BOUNCING BALLS we set $n$ to 8, 16, and 32, respectively.

### E.4.3 ENCODER

Encoder maps observations $\boldsymbol{y}_1, ..., \boldsymbol{y}_N$ to parameters $\boldsymbol{\psi}_1, ..., \boldsymbol{\psi}_B$ of the approximate posterior (Equation 75). In particular, it returns the means $\boldsymbol{\gamma}_1, ..., \boldsymbol{\gamma}_B$ and standard deviations $\boldsymbol{\tau}_1, ..., \boldsymbol{\tau}_B$ of the normal distributions $\mathcal{N}(\boldsymbol{s}_1|\boldsymbol{\gamma}_1, \text{diag}(\boldsymbol{\tau}_1^2)), ..., \mathcal{N}(\boldsymbol{s}_B|\boldsymbol{\gamma}_B, \text{diag}(\boldsymbol{\tau}_B^2))$. Using second order dynamics naturally suggests splitting the parameters into two groups. The first group contains parameters for the "position" part of the latent space, while the second group contains parameters for the "velocity" part. So, we split the means and standard deviations into position and velocity parts as

$$\boldsymbol{\gamma}_b = \left( \begin{array}{c} \boldsymbol{\gamma}_b^{\text{p}} \\ \boldsymbol{\gamma}_b^{\text{v}} \end{array} \right), \boldsymbol{\tau}_b = \left( \begin{array}{c} \boldsymbol{\tau}_b^{\text{p}} \\ \boldsymbol{\tau}_b^{\text{v}} \end{array} \right), \quad b \in \{1, ..., B\}, \tag{78}$$

where the position and velocity parts occupy a half of the latent space each (have dimension $d/2$). Then, we simply make each $\boldsymbol{\psi}_i$ contain the means and standard deviations as:

$$\boldsymbol{\psi}_1, ..., \boldsymbol{\psi}_B = \left( \begin{array}{c} \boldsymbol{\gamma}_1^{\text{p}} \\ \boldsymbol{\tau}_1^{\text{p}} \\ \boldsymbol{\gamma}_1^{\text{v}} \\ \boldsymbol{\tau}_1^{\text{v}} \end{array} \right), ..., \left( \begin{array}{c} \boldsymbol{\gamma}_B^{\text{p}} \\ \boldsymbol{\tau}_B^{\text{p}} \\ \boldsymbol{\gamma}_B^{\text{v}} \\ \boldsymbol{\tau}_B^{\text{v}} \end{array} \right). \tag{79}$$

$y_{1:N}$

$\downarrow h_{\text{comp}}$

$a_{1:N}$

$h_{\text{agg}}^{\text{p}} \diagdown h_{\text{agg}}^{\text{v}}$

$b_{1:B}^{\text{p}} \quad b_{1:B}^{\text{v}}$

$\searrow$

$b_{1:B}$

$\downarrow h_{\text{read}}$

$\boldsymbol{\psi}_{1:B}$

Figure 20: Encoder for 2nd order dynamics.

In Section 3.3 we described the structure of our encoder. For the ease of exposition we omitted overly general descriptions and presented a simple to understand overall architecture (Figure 5 (a)). However, in practice we use a slightly more general setup which we show in Figure 20. As can be seen, we simply use two aggregation function $h_{\text{agg}}^{\text{p}}$ and $h_{\text{agg}}^{\text{v}}$ to aggregate information for the position and velocity components separately. Then, we concatenate $\boldsymbol{b}_{1:B}^{\text{p}}$ and $\boldsymbol{b}_{1:B}^{\text{v}}$ to get $\boldsymbol{b}_{1:B}$. Other components remain exactly the same as described in Section 3.3.

Now, we describe the sub-components of the encoder:

$h_{\textbf{comp}}$ is represented as a convolutional neural network (CNN). In particular, $h_{\text{comp}}$ has the following architecture: three convolution layers (5x5 kernel, stride 2, padding 2) with batch norm and ReLU nonlinearities, one convolution layer (2x2 kernel, stride 2) with batch norm and ReLU nonlinearities, linear layer. The four convolution layers have $n$, $2n$, $4n$ and $8n$ channels, respectively. For datasets PENDULUM, RMNIST, and BOUNCING BALLS we set $n$ to 8, 16, and 32, respectively.

$h_{\mathbf{agg}}^{\mathbf{p}}$ **and** $h_{\mathbf{agg}}^{\mathbf{v}}$ are transformer encoders with our temporal dot product attention and relative positional encodings (Section 3.3). The number of layers (i.e., $L$ in Figure 5) is 4 for $h_{\text{agg}}^{\text{p}}$ and 8 for $h_{\text{agg}}^{\text{v}}$. We set $D_{\text{low}} = 128$, $\epsilon = 10^{-2}$, $p = \infty$ (i.e., use masking), and finally we set $\delta_r$ to 15% of the training time interval $[t_1, t_N]$. For both aggregation functions we use only temporal attention at the first layer since we found that it slightly improves the performance. In Appendix F we investigate the effects that $p$ and $\delta_r$ have on the model's performance.

$h_{\mathbf{read}}$ is a mapping from $\boldsymbol{b}_i$ to $\boldsymbol{\psi}_i$. Recall that we define $\boldsymbol{b}_i$ as

$$\boldsymbol{b}_i = \begin{pmatrix} \boldsymbol{b}_i^{\text{p}} \\ \boldsymbol{b}_i^{\text{v}} \end{pmatrix}, \tag{80}$$

so $h_{\text{read}}$ is defined as

$$h_{\text{read}}(\boldsymbol{b}_i) = \begin{pmatrix} \text{Linear}(\boldsymbol{b}_i^{\text{p}}) \\ \exp\left(\text{Linear}(\boldsymbol{b}_i^{\text{p}})\right) \\ \text{Linear}(\boldsymbol{b}_i^{\text{v}}) \\ \exp\left(\text{Linear}(\boldsymbol{b}_i^{\text{v}})\right) \end{pmatrix} = \begin{pmatrix} \boldsymbol{\gamma}_i^{\text{p}} \\ \boldsymbol{\tau}_i^{\text{p}} \\ \boldsymbol{\gamma}_i^{\text{v}} \\ \boldsymbol{\tau}_i^{\text{v}} \end{pmatrix} = \boldsymbol{\psi}_i, \tag{81}$$

where $\text{Linear}()$ is a linear layer (different for each line).

**Constraining variance of the approximate posteriors** As we showed is Section 4.4, forcing the variance of the approximate posteriors $q_{\boldsymbol{\psi}_i}(\boldsymbol{s}_i)$ to be at least $\tau_{\text{min}}^2 > 0$ in each direction might greatly improve the model's performance. In practice, we implement this constraint by simply adding $\tau_{\text{min}}$ to $\tau_i^{\text{p}}$. We do not add $\tau_{\text{min}}$ to $\tau_i^{\text{v}}$ as we found that it tends to make long-term predictions less accurate.

**Structured Attention Dropout** We found that dropping the attention between random elements of the input and output sequences improves performance of our model on regular time grids and for block sizes larger than one. In particular, at each attention layer we set an element of the unnormalized attention matrix $\boldsymbol{C}_{ij}^{\text{DP}} + \boldsymbol{C}_{ij}^{\text{TA}}$ to $-\infty$ with some probability (0.1 in this work). This ensures that the corresponding element of $\boldsymbol{C}_{ij}$ is zero. This is similar to DropAttention of Zehui et al. (2019), however in our case we do not drop arbitrary elements, but leave the diagonal of $\boldsymbol{C}_{ij}$ and one of the first off diagonal elements unchanged. This is done to ensure that the output element $i$ has access to at least the $i$'th element of the input sequence and to one of its immediate neighbors.

## F PROPERTIES OF THE ENCODER

Our encoder has parameters $p$ and $\delta_r$ which control the shape and size, respectively, of the temporal attention windows (see Section 3.3). Here we investigate how these parameters affect our model's performance. At test time we assume to have access to observations within some initial time interval $[t_1, t_1 + t_{\text{test}}]$. Figure 21 (left) shows that there seems to be no conclusive effect from the shape of the attention window. On the other hand, as Figure 21 (right) shows, parameter $\delta_r$ seems to have noticeable effect on all three datasets. We see that the curves have the U-shape with the best performance being at $\delta_r = \delta_{\text{test}}/2$. We also see that too wide attention windows (i.e., large $\delta_r$) tend to increase the error.

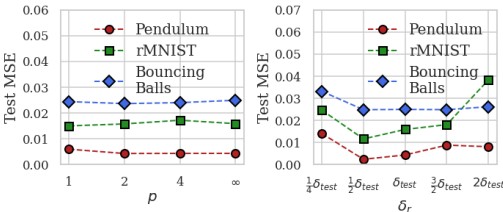

Figure 21: Test errors for different values of $p$ and $\delta_r$

# G   COMMON HEURISTICS

Here we provide details about our heuristics comparison setup in Section 4.5.

## G.1   SETUP

In all cases (SS, SS+sub, SS+progr, Ours), training, testing and model setups are described in Appendix E. The only difference between the single shooting version of our model (SS) and the multiple shooting version (Ours) is the number of blocks. For SS we use a single block, while for Ours we use multiple blocks (see Appendix E).

## G.2   HEURISTICS

**Training on sub-trajectories.**   Here, instead of training on full trajectories, at each training iteration we randomly select a short sub-trajectory from each full trajectory and train on these sub-trajectories. For PENDULUM/RMNIST/BOUNCING BALLS datasets we used sub-trajectories of length 2/2/6. These sub-trajectory lengths were selected such that they are identical to the sub-trajectories used in the multiple shooting version of our model (Ours).

**Increasing training trajectory length.**   Here, instead of starting training on full trajectories, we start training on a small number of initial observations, and then gradually increase the training trajectory length. In particular, for PENDULUM and RMNIST datasets we start training on first 5 observations, and then double that length every 10k iterations until we reach the full length. For BOUNCING BALLS dataset we start training on first 2 observations, and then double that length every 10k iterations until we reach the full length.

# H   COMPARISON TO OTHER MODELS

Here we provide details about our model comparison setup in Section 4.6 and show predictions from different models.

## H.1   NODEP

NODEP is similar to our model in the sense that it also uses the encode-simulate-decode approach, where it takes some number of initial observations, maps them to a latent initial state, simulates the deterministic latent dynamics, and then maps the latent trajectory to the observation space via a decoder. The encoder works by concatenating the initial observations and their temporal positions, mapping each pair to a representation space and averaging the individual representations to compute the aggregated representation from which the initial latent state is obtained. This encoder allows NODEP to operate on irregular time grids, but, due to its simplicity (it is roughly equivalent to a single attention layer), might be unable to accurately estimate the latent initial state.

NODEP reported results on a variant of RMNSIT dataset, so we use their setup directly with our RMNIST and PENDULUM datasets. For our BOUNCING BALLS dataset we used 32 filters for the encoder and decoder (close to our model), and the same dynamics function as for our model.

We train NODEP using random subsets of the first 8 observations to infer the latent initial state. We found this approach to generalize better than training strictly on the first 8 observations. For validation and testing we always use the first 8 observations.

## H.2   ODE2VAE

ODE2VAE is similar to our model in the sense that it also uses the encode-simulate-decode approach, where it takes some number of initial observations, maps them to a latent initial state, simulates the deterministic second-order latent dynamics, and then maps the latent trajectory to the observation space via a decoder. The encoder computes the latent initial state by stacking the initial observations and passing them thought a CNN. This encoder is flexible, but restricted to regular time grids and a constant number of initial observations.

ODE2VAE reported results on variants of RMNIST and Bouncing Balls datasets, so we use their setup directly with our RMNIST and BOUNCIGN BALLS datasets. For our PENDULUM we use ODE2VAE with the same setup as for RMNIST. We tried to increase the sizes of the ODE2VAE components, but it resulted in extremely long training times.

For training, validation and testing we use the first 8 observations to infer the latent initial state.

### H.3 OUR MODEL

Our model followed the same setup as described in Appendix E.

### H.4 MORE PREDICTIONS

In the model comparison experiment (Section 4.6) we showed only the median test predictions. Here, we plot test predictions corresponding to different percentiles. Figures 22, 23, and 24 show predictions of NODEP, ODE2VAE, and our model.

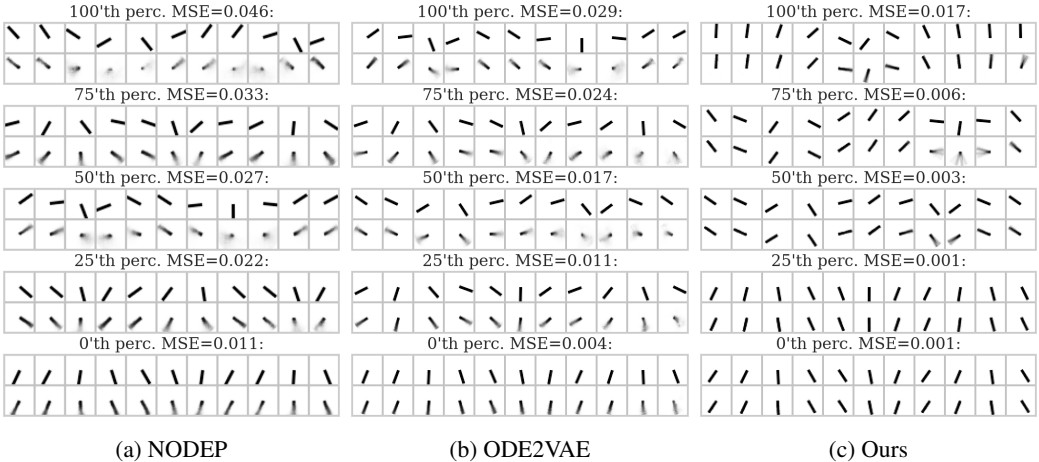

(a) NODEP   (b) ODE2VAE   (c) Ours

Figure 22: Predictions on PENDULUM dataset. Shown are test predictions corresponding to different percentiles wrt test MSE. The first snapshot is at $t_1$, the last one is at $t_{51}$. The distance between snapshots is five time points. First row is ground truth, second row is the prediction.

(a) NODEP   (b) ODE2VAE   (c) Ours

Figure 23: Predictions on RMNIST dataset. Shown are test predictions corresponding to different percentiles wrt test MSE. The first snapshot is at $t_1$, the last one is at $t_{51}$. The distance between snapshots is five time points. First row is ground truth, second row is the prediction.

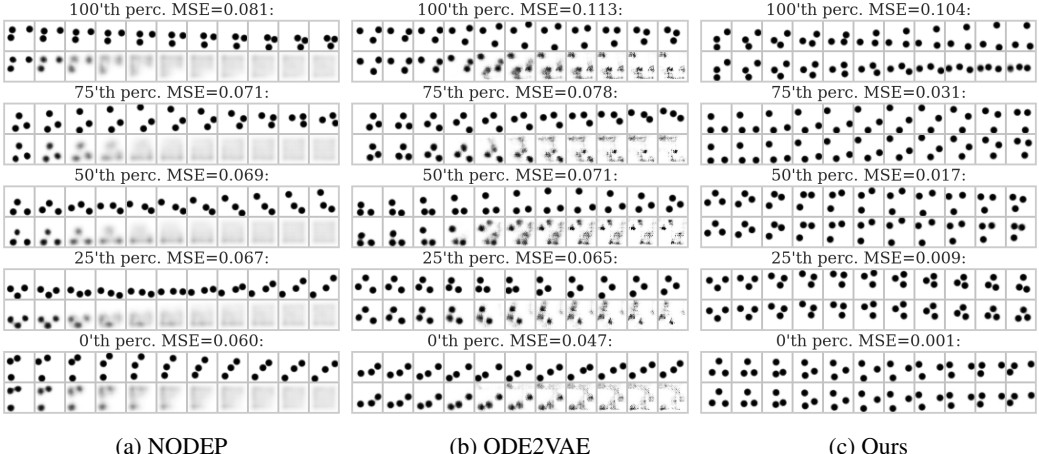

|            |            |            |
|:----------:|:----------:|:----------:|
| (a) NODEP | (b) ODE2VAE | (c) Ours |

Figure 24: Predictions on BOUNCING BALLS dataset. Shown are test predictions corresponding to different percentiles wrt test MSE. The first snapshot is at $t_1$, the last one is at $t_{51}$. The distance between snapshots is five time points. First row is ground truth, second row is the prediction.

### H.5    TRAINING WITH DIFFERENT SUB-TRAJECTORY LENGTHS

We train our model on full trajectories. Other models are trained on sub-trajectories of length $N$. Note that in this experiment we remove the iterative training heuristic from ODE2VAE to study the sub-trajectory length effects directly. All models are tested on full trajectories and use the first 8 observations to infer the latent initial state. Figure 25 shows results for different values of $N$. We see that our model outperforms NODEP and ODE2VAE in all cases. We also see that both NODEP and ODE2VAE perform poorly when trained on short sub-trajectories; in figures below we show that for $N = 10$ both models perform well on the first $N$ time points, but fail to generalize far beyond the training time intervals, which is in contrast to our model which shows excellent generalization. Increasing the sub-trajectory length tends to provide some improvement, but only up to a certain point, where the training starts to fail; in figures below we show how NODEP and ODE2VAE fail for large $N$.

Figures 26, 27, and 28 show predictions of NODEP and ODE2VAE trained on sub-trajectories of different lengths.

Overall, we see that NODEP and ODE2VAE tend to perform well when trained and tested on short trajectories, but do not generalize beyond the training time interval very well. Simply training these models on longer sequences does not necessarily help as the optimization problem becomes harder and training might fail. Our model provides a principled solution to this dilemma by splitting long trajectories into short blocks and utilizing the continuity prior to enforce consistency of the solution across the blocks thus ensuring easy and fast training with stable predictions over long time intervals.

### H.6    COMPARISON AGAINST ANOTHER MULTIPLE-SHOOTING-BASED METHOD

We compare the performance of our method against Jordana et al. (2021) which use a deterministic discrete-time latent dynamics model and apply multiple shooting directly in the latent space without amortization. After training the model, the optimized shooting variables are used to train a discrete-time RNN-based recognition network to map observations to the corresponding shooting variables. The recognition network is then used at test time to map initial observations to the latent initial state.

We use the official implementation from Jordana et al. (2021). For PENDU-LUM/RMNIST/BOUNCIGN BALLS datasets we use the penalty constant of 1e3/1e3/1e4, learning rate of 1e-3/1e-3/3e-4, batch size of 16/16/64, number of training epochs of 600/600/3000. In all cases the number of shooting variables is set to 5.

In all cases, architecture of the dynamics function and decoder is the same as for our model. The encoder of Jordana et al. (2021) first maps the images to low-dimensional vectors using a CNN (we

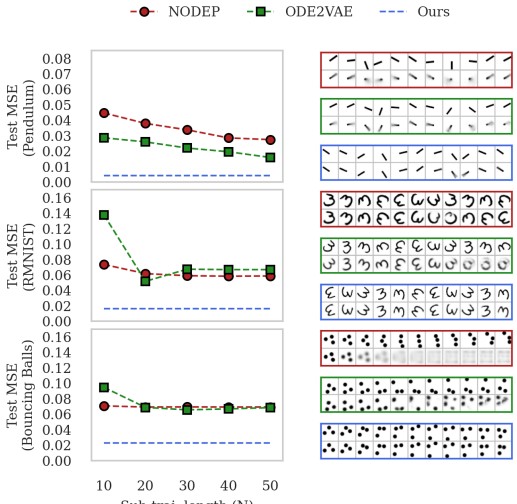

Figure 25: **Left**: Test errors for different models and datasets. **Right**: For each dataset, we plot ground truth and predictions for NODEP, ODE2VAE and our model (top to bottom). Each sub-plot shows the ground truth as the first row, and the prediction as the second row. We plot test prediction with the median test error (for each model and dataset we select the value of $N$ which gives the best predictions).

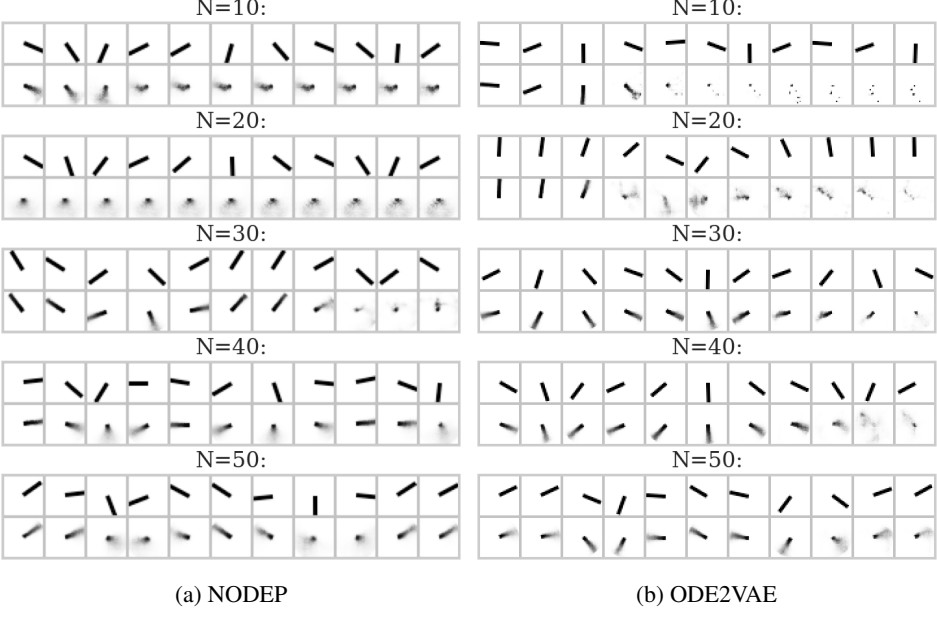

Figure 26: Predictions of NODEP and ODE2VAE on PENDULUM dataset when trained of sub-trajectories of length $N$. Shown are test predictions with the median test error. The first snapshot is at $t_1$, the last one is at $t_{51}$. The distance between snapshots is five time points. First row is ground truth, second row is the prediction.

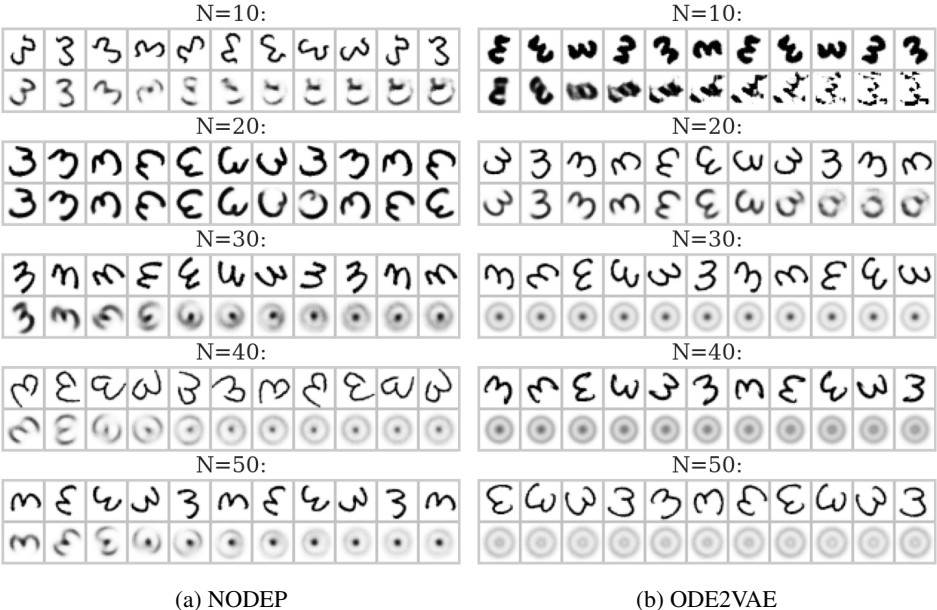

(a) NODEP

(b) ODE2VAE

Figure 27: Predictions of NODEP and ODE2VAE on RMNIST dataset when trained of sub-trajectories of length $N$. Shown are test predictions with the median test error. The first snapshot is at $t_1$, the last one is at $t_{51}$. The distance between snapshots is five time points. First row is ground truth, second row is the prediction.

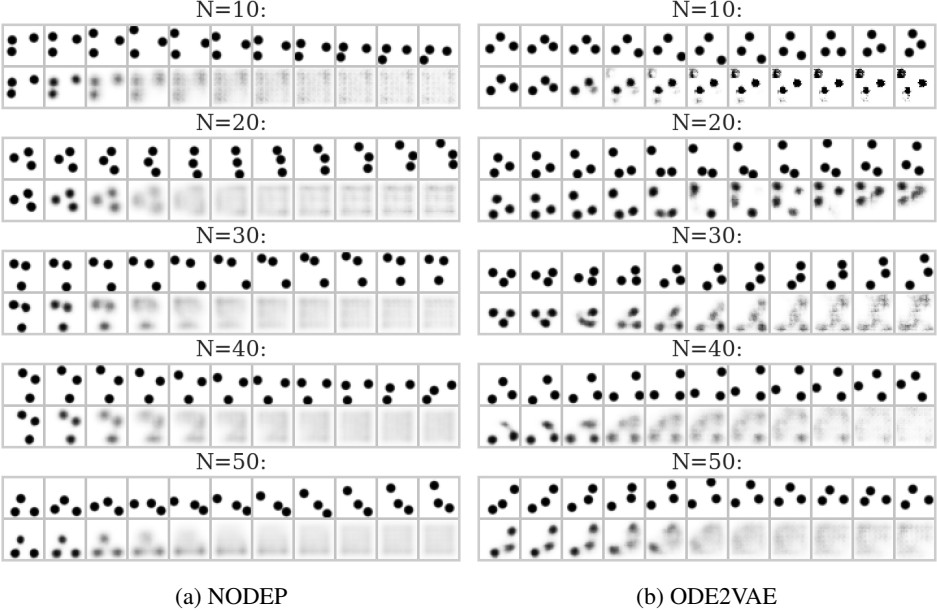

(a) NODEP

(b) ODE2VAE

Figure 28: Predictions of NODEP and ODE2VAE on BOUNCING BALLS dataset when trained of sub-trajectories of length $N$. Shown are test predictions with the median test error. The first snapshot is at $t_1$, the last one is at $t_{51}$. The distance between snapshots is five time points. First row is ground truth, second row is the prediction.

| Dataset | Test MSE (Ours) | Test MSE (Jordana et al. (2021)) |
|---|---|---|
| Pendulum (reg.) | 0.004 | 0.005 |
| RMNIST (reg.) | 0.016 | 0.020 |
| Bouncing Balls (reg.) | 0.023 | 0.081 |
| Pendulum (irreg.) | 0.004 | 0.029 |
| RMNIST (irreg.) | 0.015 | 0.072 |
| Bouncing Balls (irreg.) | 0.024 | 0.096 |

Table 2: Comparison results.

used the same architecture as for our model), and then applies an LSTM (we used the latent state of dimension $1024$) to map these vectors to shooting variables. Note that the encoder is trained after the model. The latent space dimension is the same as for our model. At test time we use the first $8$ observations to infer the latent initial state.

We applied the method of Jordana et al. (2021) on our datasets with regular and irregular time grids and report the results in Table 2. We found that Jordana et al. (2021) performs quite similarly to our method on regularly sampled PENDULUM and RMNIST datasets, but fails to produce stable long-term predictions on the BOUNCIGN BALLS dataset. Also, due to being a discrete-time method, Jordana et al. (2021) fails on irregularly sampled versions of the datasets.

## I  STRENGTH OF THE CONTINUITY PRIOR VS GAP BETWEEN BLOCKS

We investigate how the strength of the continuity prior (as measured by $\sigma_c$) affects the gap between consecutive blocks of the latent trajectory. We train our model with different values of $\sigma_c$ and compute the mean squared gap between the end of a current block and the beginning the next block (i.e., between the latent state $x$ at a time $t_{[b]}$ and the shooting variable $s_{[b]}$). We report the results in Table 3. We see that stronger continuity prior (i.e., smaller $\sigma_c$) tends to result in smaller gap between the blocks and, consequently, in better continuity of the whole trajectory. We also see that better continuity tends to result in smaller prediction errors.

| $\sigma_c$ | Pendulum | | RMNIST | | Bouncing Balls | |
|---|---|---|---|---|---|---|
| | Test MSE | Avg. gap | Test MSE | Avg. gap | Test MSE | Avg. gap |
| 2e-1 | 0.189 | 1.3223 | 0.104 | 6.2465 | 0.0805 | 0.0929 |
| 2e-2 | 0.028 | 0.0326 | 0.062 | 0.5094 | 0.0724 | 0.0849 |
| 2e-3 | 0.012 | 0.0017 | 0.027 | 0.0101 | 0.0475 | 0.0121 |
| 2e-4 | 0.002 | 0.0004 | 0.017 | 0.0009 | 0.0243 | 0.0012 |
| 2e-5 | 0.004 | 0.0004 | 0.015 | 0.0004 | 0.0825 | 0.0002 |

Table 3: Dependence of test MSE and inter-block continuity on $\sigma_c$.

## J  USING ODE-RNN AS AGGREGATION FUNCTION

Here we test the effect of replacing our transformer-based aggregation function $h_{\text{agg}}$ by ODE-RNN (Rubanova et al., 2019). For each dataset, we set ODE-RNN's hyperparameters such that the number of parameters is similar to that of our transformer-based $h_{\text{agg}}$. We report the results in Table 4. We see that on the PENDULUM dataset ODE-RNN works on par with our method, while on other datasets it has higher test error. The training time for ODE-RNN tends to be much larger than for our method highlighting the effectiveness of parallelization provided by the Transformer architecture.

| Dataset | Test MSE (Ours) | Test MSE (ODERNN) | Training time (Ours) | Training time (ODERNN) |
|---|---|---|---|---|
| Pendulum | 0.004 | 0.007 | 5 hours | 68 hours |
| RMNIST | 0.015 | 0.027 | 6 hours | 98 hours |
| Bouncing Balls | 0.024 | 0.036 | 34 hours | 133 hours* |

Table 4: Test MSE and training times for transformer-based and RNN-based aggregation functions. *Trained with block size of 1 due to long training times.

