# OpenReview forum: "Latent Neural ODEs with Sparse Bayesian Multiple Shooting"
_ICLR.cc/2023/Conference — ICLR 2023 poster_

### Official Review · Reviewer_9Vai · 2022-10-16

**Confidence:** 4
**Correctness:** 4
**Technical Novelty And Significance:** 3
**Empirical Novelty And Significance:** 3
**Recommendation:** 8

**Clarity, Quality, Novelty And Reproducibility:**

The paper is clear to read, understand, and easy to follow. The novelty of the paper might not be high as based on the authors, there have been similar works of using MS in NODEs. However, the quality of the paper is good as they have built a unified probabilistic framework to incorporate the ideas from latent ODEs and MS to tackle the problem of learning in long trajectories.

**Strength And Weaknesses:**

Strength:

- Improved generalization and test error.
- Improved training time.

- Loss function has a lot of terms which can make finding the optimal wighting of terms more difficult.
- Lack of evaluation based on RNN encoders.

**Summary Of The Paper:**

Neural ODEs (NODE) are very powerful to model the dynamical systems. In particular, latent NODE models are great candidate when we deal with very high-dimensional data, as they learn the dynamics in a much lower dimensional space. This paper (along with some other papers they have cited) claims that when the length of trajectories (time span) starts to grow, the loss landscape of the NODE models become very complicated leading to poor estimates and generalization. To solve this problem, they employ a technique from optimal control called "multiple shooting" (MS), which works has follows. The authors chop the whole trajectory to multiple blocks such that each block contains fewer observation. This allows learning ODEs for smaller periods of time and also parallelization which makes the training much faster. There is a risk that smaller blocks might be disconnected. In order to tackle this issue, they add a continuity prior in a Bayesian framework which also learns a sparse blocks. Overall, the results show that this method is effective in both improving the test errors and decreasing the training time.

**Summary Of The Review:**

The paper is well written and clear. I am mostly convinced that this method is helpful in learning the dynamics for the long trajectories and appreciate the innovative solutions like MS and using Bayesian framework to enable continuity prior and uncertainty of the dynamics. However, I have some questions:

- I wonder where the complexity of long trajectories come from? Is it because of error accumulation on ODE solvers for long times? If this is the case, can we shrink the time span by preserving the ordering of the observation over time? For example, if the original time span is [0,50] second, can we scale it to [0,5] and then solve the ODEs, or will we still see the same problem?

- The new loss function proposed by the authors in (17-19) has six different terms. Usually, in VAE models, the optimal performance is achieved by some non-uniform weights of these terms. I wonder if the authors have played with weighting these terms? Have the authors found that optimizing the loss as is in (17-19) gives the best results? I feel this complicated loss should be tricky to train without knowing the optimal weights and curious to see how that is the case with the authors' experience.

- Why have the authors chosen to use transformer instead of RNNs in the encoder? The original work of latent ODE uses RNN and also consider the case of irregular time-series. I wonder if the authors have tried RNN as well and not found it satisfying? I would recommend adding it as another comparison so that we can see how much adding transformers help in encoders.

---

> ### Author Response · Authors · 2022-11-15
> **Response**
>
> We thank the reviewer for their thoughtful comments and constructive suggestions.
>
> >**Q1:** *"I wonder where the complexity of long trajectories come from? Is it because of error accumulation on ODE solvers for long times? If this is the case, can we shrink the time span by preserving the ordering of the observation over time? For example, if the original time span is [0,50] second, can we scale it to [0,5] and then solve the ODEs, or will we still see the same problem?"*
> **A1:** This is an interesting question. To the best of our understanding, errors introduced by an ODE solver is not what makes optimization with long trajectories complex. As was shown in [2], complexity (Lipschitz constant) of the loss function grows with the trajectory length (T) and Lipschitz constant (L) of the transition function. Scaling T (e.g., from 10 to 1) reduces the complexity. On the other hand, L will grow as the transition function now has to accommodate all changes in the data in a much shorter time interval. Hence, the overall effect of scaling on complexity of the optimization problem might be hard to predict.
> To obtain some empirical results, we repeated the experiment in Figure 1 using both the original time grid [0, 20] and its scaled version [0, 2]. We observed that in both cases the loss complexity and its growth with the trajectory length N are very similar. We also observed that convergence with the scaled time grid is slower, so we ran the optimization for 9000 iterations per cycle (instead of 3000 for the original time grid).
>
> >**Q2:** *"The new loss function proposed by the authors in (17-19) has six different terms. Usually, in VAE models, the optimal performance is achieved by some non-uniform weights of these terms. I wonder if the authors have played with weighting these terms? Have the authors found that optimizing the loss as is in (17-19) gives the best results? I feel this complicated loss should be tricky to train without knowing the optimal weights and curious to see how that is the case with the authors' experience."*
> **A2:** We observed that under some hyper-parameter configurations the continuity-promoting term (iv) might cause the shooting variables to collapse to a single point hence preventing the learning of meaningful dynamics. Downscaling this term helps to avoid the collapse. However, in our experiments we did not use any scaling and optimized the loss function in Eqs. 17-19 directly. We have added this discussion to Appendix C1.
>
> >**Q3:** *"Why have the authors chosen to use transformer instead of RNNs in the encoder? The original work of latent ODE uses RNN and also consider the case of irregular time-series. I wonder if the authors have tried RNN as well and not found it satisfying? I would recommend adding it as another comparison so that we can see how much adding transformers help in encoders."*
> **A3:** The main reasons why we built our encoder using transformer instead of ODERNN [1] are parallelization and simplicity of including temporal inductive biases. Transformer computes all variational parameters in parallel thus providing noticeable computation speed improvements over ODERNN. Also, making transformer work on irregular time grids and endowing it with appropriate inductive biases (such as temporal locality and independence on the global time) was possible and did not require fundamental changes to the original transformer architecture.
> We replaced our transformer with ODERNN and report the results below.
> >>*Implementation:* We use the official implementation of ODERNN from [1] and replace our transformer-based aggregation function h_agg (Figure 5) by ODERNN.
> *Experiment:* Training on Pendulum dataset.
> *Results:* We found that on the Pendulum dataset ODERNN works well and achieves the MSE value of 0.007 which is close to the MSE of our transformer-based aggregation model (0.004). However, ODERNN requires much longer training time due to ODERNN being much slower than our transformer.  Because of long training times we do not yet have results for RMNIST and Bouncing Balls datasets. We will also run this experiment on RMNIST and Bouncing Balls datasets and add the results to the revised version of our manuscript. We summarize the results in the table below.
> | Dataset        | Test MSE (Ours) | Test MSE (ODERNN) | Training time (Ours) | Training time (ODERNN) |
> |----------------|:---------------:|:-----------------:|:--------------------:|:----------------------:|
> | Pendulum       |       0.004      |        0.007       |         4.3 hours         |          39.2 hours          |
>
> **References:**
> [1] Rubanova et al., "Latent odes for irregularly-sampled time series", NeurIPS, 2019
> [2] Ribeiro et al., "On the smoothness of nonlinear system identification"

---

> > ### Comment · Reviewer_9Vai · 2022-11-23
> > **Thanks for the response**
> >
> > I would like to thank the authors for addressing my comments and questions. Great work!

---

### Official Review · Reviewer_AJZS · 2022-10-19

**Confidence:** 4
**Correctness:** 4
**Technical Novelty And Significance:** 4
**Empirical Novelty And Significance:** 4
**Recommendation:** 10

**Clarity, Quality, Novelty And Reproducibility:**

The work is extremely clear, and the writing is very high-quality.

The work is high-quality and novel.

There are sufficient details provided to reproduce the experiments.


**Strength And Weaknesses:**

Overall, the paper is very strong.

The model is clearly explained, both on a theoretical level, and how to make it work in practice. There are thorough ablations and comparisons to other methods.

I only have one question/concern:

How continuous are the models in practice? Does better continuity correlate with better performance, or is the model learning to make it discontinuous and then learning that?
I know there is an experiment showing this for different strengths of the continuity prior, but I’m wondering if you have an experiment that measures continuity itself directly, rather than a parameter that is supposed to encourage it.
How correlated is the continuity prior with actual continuity (e.g. MSE between shooting points and trajectory points)?


**Summary Of The Paper:**

This work tackles the issue of modelling time-series data tractably and efficiently. Modelling long, irregular time sequences can be very challenging to optimise (in terms of the underlying landscape of the loss function), and inefficient (simulation of the time series in serial can be very slow). This work adapts the well-known technique of multiple-shooting, but reframes with a Bayesian lens. This allows modelling a time series as a series of blocks that can be simulated in parallel, which resolves both issues of efficiency and optimisation.

A variational framework is used. This allows factorising the shooting variables so they can be sampled independently (hence in parallel).

For the encoder, the authors develop a novel attention mechanism and positional encoding.

**Summary Of The Review:**

Overall, the paper is very solid. The method is novel and interesting, and thorough ablations and comparisons are done to a high-standard with plenty of information for reproducibility. I have a few questions (listed in Strengths and Weaknesses), but overall I’m happy to recommend this paper for acceptance.

---

> ### Author Response · Authors · 2022-11-15
> **Response**
>
> We thank the reviewer for their thoughtful comments and constructive suggestions.
>
> **Q1:** *"How continuous are the models in practice? Does better continuity correlate with better performance, or is the model learning to make it discontinuous and then learning that? I know there is an experiment showing this for different strengths of the continuity prior, but I’m wondering if you have an experiment that measures continuity itself directly, rather than a parameter that is supposed to encourage it. How correlated is the continuity prior with actual continuity (e.g. MSE between shooting points and trajectory points)?"*
> **A1:** This is an interesting question. We did as you suggested and trained our model with different values of $\sigma_c$ and then measured the average gap (MSE) between sub-trajectories. We conducted this experiment on the Pendulum dataset and report the results below.
>
> | $\sigma_c$ | Test MSE | Avg. gap (MSE) |
> |:-------:|:--------:|:--------:|
> |   2e-1  |   0.189   |   1.3223   |
> |   2e-2  |   0.028   |   0.0326   |
> |   2e-3  |   0.012   |   0.0017   |
> |   2e-4  |   0.002   |   0.0004   |
> |   2e-5  |   0.004   |   0.0004   |
>
> We see that stronger continuity prior (i.e., smaller $\sigma_c$) results in smaller gap between sub-trajectories and, consequently, in better continuity of the whole trajectory. We also see that better continuity tends to result in smaller prediction errors. We will conduct this experiment on RMNIST and Bouncing Balls datasets and add the results to the revised manuscript.

---

> > ### Comment · Reviewer_AJZS · 2022-11-15
> > **Response**
> >
> > Thanks for your response, and including the additional result. It's very interesting.
> >
> > I'm happy to increase my score, really excellent work!

---

### Official Review · Reviewer_Lhfw · 2022-10-24

**Confidence:** 4
**Clarity, Quality, Novelty And Reproducibility:** See above.
**Correctness:** 3
**Technical Novelty And Significance:** 2
**Empirical Novelty And Significance:** 2
**Recommendation:** 6

**Strength And Weaknesses:**

**Strengths**
- The paper tackles an important challenge of training models given long and irregularly sampled datasets
- The paper is well-written and easy to follow
- The proposed approach seems easy to implement

**Weakness**

*Technical Limitations*
-  The novelty is limited. The proposed approach seems to be a straightforward application of amortized inference and attention
- The discussion of the proposed temporal attention needs to be expanded. There are several approaches for temporal encoding. Did the paper consider alternative formulations to Eqn. 23?
-  How is the block size chosen given observational data?
- The performance gains seem marginal when compared to other models such as ODE2VAE and NODEP
-  Eqn 17: What is the assumed data likelihood? It's unclear why the data likelihood depends on $s_1$ only.
- Overall the paper proposes several heuristics for training without proper justification, *e.g.*, the paper claims constraining posterior variance is crucial without providing specific details of the actual approach. Hence, it's unclear where the performance benefits come from.

*Experimental Evaluation*
- The paper needs to discuss and compare with other latent-ODEs approaches for irregularly sampled observations, *e.g.*, [1, 2] and temporal attention mechanisms [3]
- The paper evaluates relatively clean synthetic datasets. The evaluation should also consider challenging real-world datasets such as Physionet

**References**
- [1]  De Brouwer et al., "Gru-ode-bayes: Continuous modeling of sporadically-observed time series",  NeurIPS, 2019
- [2] Rubanova et al.,  "Latent odes for irregularly-sampled time series", NeurIPS, 2019
- [3] Zhang et al., "Self-attentive Hawkes process", ICML, 2020

**Summary Of The Paper:**

The paper proposes an amortized inference approach combined with a temporal attention mechanism for parallel training of dynamical models given long irregularly sampled trajectories. Experimental results on three synthetic datasets demonstrate improved mean squared error compared to baselines.

**Summary Of The Review:**

The technical novelty is limited. Additionally, experimental evaluations are missing several baselines and rely on clean synthetic datasets. Also, the paper proposes several training heuristics with little justification. Hence the source of performance gain is unclear.

---

> ### Author Response · Authors · 2022-11-15
> **Response 1/2**
>
> We thank the reviewer for their thoughtful comments and constructive suggestions.
>
> > **Q1:** *"The novelty is limited. The proposed approach seems to be a straightforward application of amortized inference and attention."*
> **A1:** Using amortized inference and attention is definitely not novel. The main novelties of our work are: 1) development of a method for efficient, quick, and stable training of dynamic models on long trajectories; and 2) modifications to the attention mechanism which make it applicable on irregular time grids.
>
> >**Q2:** *"The discussion of the proposed temporal attention needs to be expanded. There are several approaches for temporal encoding. Did the paper consider alternative formulations to Eqn. 23?"*
> **A2:** We believe the current version of the text contains all details and motivation for *what* our temporal attention does, but we would be happy to expand it if something is unclear, or some information is missing.
> However, we agree that our manuscript does not give a clear motivation for *why* this particular form of temporal attention was selected. The reason is quite straightforward - we needed a simple mechanism for controlling the shape of attention windows while adding minimal modifications to the original dot-product attention. The current form of temporal attention (Eq. 23) had all these properties and showed good performance, so we used it.
> Alternative (simpler) versions of Eq. 23 were considered, but ultimately discarded due to poor performance or insufficient control over the attention windows.
>
> >**Q3:** *"How is the block size chosen given observational data?"*
> **A3:** Block size is a hyper-parameter, so it can be selected using any model selection technique. We recommend starting with the block size of 1 and slowly increasing it (as in Fig. 9).
>
> >**Q4:** *"The performance gains seem marginal when compared to other models such as ODE2VAE and NODEP."*
> **A4:** We respectfully disagree. We provide Fig. 13 with quantitative (MSE) and qualitative (visualizations) results which show that our method performs significantly better (has 3-5 times smaller MSE and better visual quality) than ODE2VAE and NODEP. We further provide Figs. 22-24 for more detailed MSE/visual comparisons.
>
> >**Q5:** *"Eqn 17: What is the assumed data likelihood? It's unclear why the data likelihood depends on s_1 only."*
> **A5:** We assume Gaussian likelihood. The first term (i) contains only the first observation y_1, so it depends only on s_1. All following observations are contained in term (ii) which depends on all shooting variables. We provide all details about the model in Appendix B (see Eqs. 30-33).
>
> >**Q6:** *"Overall the paper proposes several heuristics for training without proper justification, e.g., the paper claims constraining posterior variance is crucial without providing specific details of the actual approach. Hence, it's unclear where the performance benefits come from."*
> **A6:** This is a good point. Constraining the posterior variance is useful, but definitely not crucial. Results in Fig. 11, show that our model achieves good performance and outperforms the baselines even without the constraint. We also provide all details about the implementation in Appendix E.4.3.
>
> >**Q7:** *"The paper needs to discuss and compare with other latent-ODEs approaches for irregularly sampled observations, e.g., [1, 2] and temporal attention mechanisms [2]"*
> **A7:**
> >>The method of [2] uses dot-product attention with global positional encodings. As we showed in Table 1, this setup works poorly in our case. We added [2] as a related work.
>
> >>***Latent ODE + ODERNN [1]:***
> *Implementation:* We implemented Latent ODE + ODERNN using the official implementation of ODERNN from [1]. To adapt [1] to our setting, we compress the observations using a CNN before applying ODERNN. The dynamics function and decoder are the same as for our model.
> *Experiment:* Training on the Pendulum dataset.
> *Results:* Being a single shooting model, [1] suffers from the long trajectory problem and does not support parallelization of the ODE solver which results in poor performance and long training times. Due to long training times we trained [1] only for 95k iterations. We summarize the results in the table below.
> | Dataset  | Test MSE (Ours) | Test MSE ([1]) | Training time (Ours) | Training time ([1]) |
> |----------|:---------------:|:--------------:|:--------------------:|:-------------------:|
> | Pendulum |       0.004      |      0.034      |         4.3 hours         |         24 hours        |

---

> > ### Author Response · Authors · 2022-11-15
> > **Response 2/2**
> >
> > >**Q8:** *"The paper evaluates relatively clean synthetic datasets. The evaluation should also consider challenging real-world datasets such as Physionet."*
> > **A8:** Real-world datasets could be an interesting addition to our work. However, the datasets we used are common benchmarks that posses all required properties (have irregular time grids and cause the long trajectory problem). Hence, they are sufficient to demonstrate the main claims of our paper (that our method works on irregular time grids and solves the long trajectory problem).
> >
> > **References:**
> > [1] Rubanova et al., "Latent odes for irregularly-sampled time series", NeurIPS, 2019
> > [2] Zhang et al., "Self-attentive Hawkes process", ICML, 2020

---

### Official Review · Reviewer_XZSe · 2022-10-25

**Confidence:** 3
**Correctness:** 3
**Technical Novelty And Significance:** 3
**Empirical Novelty And Significance:** Not applicable
**Recommendation:** 6

**Clarity, Quality, Novelty And Reproducibility:**

**Novelty:** The paper presents a novel approach involving Beyesian inference for a quick and stable training of latent ODE models on long trajectories.  It also introduces a transformer-based encoder with novel time-attention and relative positional embeddings for irregulalry sampled data.

**Reproducibility:** Not reproducible. Although the paper includes the hyperparameters, the code to reproduce the results is missing.

**Clarity:** The paper is well written and easy to follow.

**Strength And Weaknesses:**

### Strengths
1. The paper presents a novel approach involving Beyesian inference for a quick and stable training of latent ODE models on long trajectories. It also introduces a transformer-based encoder with novel time-attention and relative positional embeddings for irregulalry sampled data.
2. The motivation of the paper is clear and it focuses on solving an important problem. The experimental results indeed show that multiple shooting is an interesting and effective way to deal with long trajectories while reducing training times.
3. The paper shows ablations to show the effectiveness of the proposed time-aware attention and relative positional encodings.

### Weaknesses
1. Different feature in irregularly sampled sequences can have observations at different times. It is not clear how does the proposed encoder takes care of this. The paper is also missing an important related work [1] in this area which proposes an attention based encoder for irregularly sampled time series.
2. Another concern with the paper is the lack of rigorous experimentation to show the usefulness of the proposed method for irregularly sampled time series datasets. The paper does not include any experiments on real-world irregularly sampled time series datasets (e.g. PhysioNet, MIMIC-III). I would expect to see at least the comparison on the PhysioNet experiments form Latent ODE paper.
3. The paper defines short trajectory with N = 10 in Figure 1 but the Latent ODE is running in continuous time. N = 10 can constitute of much longer trajectory based on the time difference between the observations or a much shorter if the observations are close to each other in time.
4. The paper is missing important comparisons with the Latent ODE work with ODERNN encoder.
5. How does the block size impact the MSE and training time on irregularly sampled time series? Empirical study with PhysioNet or MIMIC-III/IV would be very useful here.
6. The paper is also missing comparisons with Hedge et al(2022) or Jordana et al(2022) to show the effectiveness of their proposed bayesian multiple shooting approach.

[1] Shukla, Satya Narayan, and Benjamin M. Marlin. "Multi-time attention networks for irregularly sampled time series." arXiv preprint arXiv:2101.10318 (2021).

**Summary Of The Paper:**

The paper focuses on the problem of improving neural ODEs for learning from long trajectories. It shows that the loss landscape of latent neural ODEs is adversely affected by the length of the time interval and  complexity of loss can grow dramatically with increase in the length of the trajectory. The paper proposes bayesian multiple shooting technique to address this issue. Multiple shooting technique splits the trajectory into multiple short segments which are optimized in parallel while controlling the continuity between multiple segments. Standard multiple shooting approaches enforces continuity of the entire trajectory by applying a hard constraint or penalty term while training. The paper proposes to utilize Bayesian inference and naturally encoded continuity as a prior. The papers also proposes temporal attention and relative positional encodings to build a time-aware attention based encoder to deal with irregularly sampled input data.  Experiments on multiple datasets show that the proposed approach is able to achieve better performance and that multiple shooting clearly helps improve the performance and reduce training times via parallel computations.

**Summary Of The Review:**

My recommendation for this paper is weak accept. Although the paper proposes a novel solution and shows clear motivation behind the proposed approach, rigorous experimentation is missing. I would be happy to increase my score if my concerns are addressed.

---

> ### Author Response · Authors · 2022-11-15
> **Response 1/2**
>
> We thank the reviewer for their thoughtful comments and constructive suggestions.
>
> >**Q1:** *"Different feature in irregularly sampled sequences can have observations at different times. It is not clear how does the proposed encoder takes care of this."*
> **A1:** The data collection scheme described by the reviewer would result in extremely sparse data (in the worst case, each measurement contains only a single feature value) -- analysing such data is not our goal. We assume the whole feature vector y_i is available, so our encoder does not have any design choices that explicitly address the problem of missing features. However, missing features could be tackled using standard techniques (e.g., using masking or a default value for missing features).
> The source of confusion, perhaps, was Section 3.3, where we use the term "partially observed data". By that we simply mean that y_i does not contain all information about the system's state, e.g., y_i is only the position of a pendulum, but not its velocity.
>
> >**Q2:** *"The paper is also missing an important related work [1] in this area which proposes an attention based encoder for irregularly sampled time series."*
> **A2:** Thanks for the reference! We added it to the related work. Overall, the main differences between our method and [1] is that they use global positional encodings and do not restrict the shape of attention windows, while we use relative encodings and temporal attention to control the attention windows.
>
> >**Q3:** *"Another concern with the paper is the lack of rigorous experimentation to show the usefulness of the proposed method for irregularly sampled time series datasets."*
> **A3:** This is a good point. We agree that it is important to demonstrate how our method works on regular and irregular time grids, and also show how the modifications we proposed affect the model's performance. This is exactly what we do in Section 4.1 by showing that our method works equally well on both regular and irregular time grids, and in Table 1 by showing how our modifications to the transformer architecture affect the model's performance on irregular time grids.
>
> >**Q4:** *"The paper does not include any experiments on real-world irregularly sampled time series datasets (e.g. PhysioNet, MIMIC-III)."*
> **A4:** Real-world datasets could be an interesting addition to our work. However, the datasets we used are common benchmarks that posses all required properties (have irregular time grids and cause the long trajectory problem). Hence, they are sufficient to demonstrate the main claims of our paper (that our method works on irregular time grids and solves the long trajectory problem).
>
> >**Q5:** *"The paper defines short trajectory with N = 10 in Figure 1 but the Latent ODE is running in continuous time. N = 10 can constitute of much longer trajectory based on the time difference between the observations or a much shorter if the observations are close to each other in time."*
> **A5:** Indeed, whether N=10 is "long" or "short" depends on the context and distance between observations. In the demonstration (Fig. 1, App. A) the reviewer is referring to, the interval between the observations is such that a training trajectory of length N=10 is "short" (i.e., we can fit it easily). We have clarified this point in the experiment description (Appendix A).

---

> > ### Author Response · Authors · 2022-11-15
> > **Response 2/2**
> >
> > >**Q6:** *"The paper is missing important comparisons with the Latent ODE work with ODERNN encoder [2]. The paper is also missing comparisons with Hedge et al(2022) [3] or Jordana et al(2022) [4] to show the effectiveness of their proposed bayesian multiple shooting approach."*
> > **A6:** We did not consider [2] since the models we selected (NODEP, ODE2VAE) posses the same dynamical properties as [2] (all are "single shooting" models) and their encoders were explicitly designed for high-dimensional image observations. However, for completeness we compared against [2] and report the results below.
> > >>***Latent ODE + ODERNN [2]:***
> > *Implementation:* We implemented Latent ODE + ODERNN using the official implementation of ODERNN from [2]. To adapt [2] to our setting, we compress the observations using a CNN before applying ODERNN. The dynamics function and decoder are the same as for our model.
> > *Experiment:* Training on the Pendulum dataset.
> > *Results:* Being a single shooting model, [2] suffers from the long trajectory problem and does not support parallelization of the ODE solver which results in poor performance and long training times. Due to long training times we trained [2] only for 95k iterations. We summarize the results in the table below.
> > | Dataset  | Test MSE (Ours) | Test MSE ([2]) | Training time (Ours) | Training time ([2]) |
> > |----------|:---------------:|:--------------:|:--------------------:|:-------------------:|
> > | Pendulum |       0.004      |      0.034      |         4.3 hours         |         24 hours        |
> >
> > >>The model of [3] learns GP-based dynamics in the data space and does not use amortization, hence is not suitable in our setting.
> >
> > >>Even though [4] is a discrete-time model with discrete-time encoder, we agree that it could be an interesting baseline to assess the effectiveness of our approach. We implemented [4] and report the results below.
> > ***Our method vs [4]:***
> > *Implementation:* We use the official implementation of [4] on our datasets. The hyperparameters (learning rate, epochs, RNN encoder size, shooting variables) are tuned for the best performance. The CNN encoder, decoder, and dynamics function are the same as for our model.
> > *Experiments:* Training on Pendulum/RMNIST/Bouncing Balls datasets on regular and irregular time grids.
> > *Results:* We see that [4] performs similarly to our method on regularly sampled Pendulum and RMNIST datasets, but fails to produce stable long-term predictions on the Bouncing Balls dataset. Also, since [4] is a discrete-time method, it fails on irregularly sampled versions of the datasets. We will add this experiment to the revised version of our manuscript.
> > |         Dataset        | Ours (test MSE) | Jordana [4] (test MSE) |
> > |:----------------------|:----:|:-----------:|
> > | Pendulum (reg)         | 0.004 |     0.005    |
> > | RMNIST (reg)           | 0.016 |     0.020    |
> > | Bouncing Balls (reg)   | 0.023 |     0.081    |
> > | Pendulum (irreg)       | 0.004 |     0.029    |
> > | RMNIST (irreg)         | 0.015 |     0.072    |
> > | Bouncing Balls (irreg) | 0.024 |     0.096    |
> >
> > >**Q7:** *"How does the block size impact the MSE and training time on irregularly sampled time series?"*
> > **A7:** Right, because of irregularity, the block size of, e.g., 2 could result in sub-trajectories of very different lengths. Please note that we already address this point in our experiments since they were done with irregular time grids which contain large gaps between observation times (see Fig. 19). As Fig. 9 shows, the block size has great effect on MSE and training times even on irregularly sampled data.
> >
> > **References:**
> > [1] [Multi-Time Attention Networks for Irregularly Sampled Time Series](https://arxiv.org/pdf/2101.10318.pdf)
> > [2] [Latent ODEs for Irregularly-Sampled Time Series](https://arxiv.org/pdf/1907.03907.pdf)
> > [3] [Variational multiple shooting for Bayesian ODEs with Gaussian processes](https://arxiv.org/abs/2106.10905)
> > [4] [Learning Dynamical Systems from Noisy Sensor Measurements using Multiple Shooting](https://arxiv.org/abs/2106.11712)

---

### Decision · Program_Chairs · 2023-01-20

**Decision:**

Accept: poster

**Justification For Why Not Higher Score:**

This could be a spotlight — in fact, one of the reviewers argues as such (rating the paper a 10). I think it likely falls just short of "spotlight" level, due to some of the comments from the more borderline reviews, and also as it focuses on a fairly specialized problem.

However, I would be happy to see this as a spotlight as well.

**Justification For Why Not Lower Score:**

All reviewers (at least weakly) indicated to accept the paper, with one reviewer encouraging a spotlight or oral presentation.

**Metareview: Summary, Strengths And Weaknesses:**

There is a unanimous agreement to accept the paper among all reviewers, with two reviewers very strongly supporting the work, which develops a stable approach to training latent neural ODEs on long trajectories. All reviewers praised the paper's clear writing and presentation; the two more borderline reviews had initial concerns regarding the novelty and regarding the thoroughness of the empirical comparisons. In my opinion the authors wrote clear and detailed responses to these concerns.

**Note From Pc:**

if the above contains the word "oral" or "spotlight" please see: "oral" presentation means -> notable-top-5% and "spotlight" means -> notable-top-25%. As stated in our emails, we are disassociating presentation type from AC recommendations